

# Enhancing analogy-based software cost estimation using Grey Wolf Optimization algorithm

Taghi Javdani Gandomani[1], Maedeh Dashti[1], Sadegh Ansaripour[2] and Hazura Zulzalil[3]

[1] Department of Computer Science, Faculty of Mathematical Sciences, Shahrekord University, Shahrekord, Chaharmahal and Bakhtiari, Iran
[2] Department of Computer Engineering, Aghigh University, Shahinshahr, Isfahan, Iran
[3] Department of Software Engineering and Information Systems, Universiti Putra Malaysia, Serdang, Selangor, Malaysia

Corresponding author
Hazura Zulzalil,
hazura@upm.edu.my

## ABSTRACT

Accurate software cost estimation (SCE) is a critical factor in the successful delivery of software projects, as highlighted by industry statistics indicating that only some of the projects comply with the predicted budget. Among the software estimation methods, analogy-based estimation (ABE) is one of the most popular ones. Although this method has been customized in recent years with the help of optimization algorithms to achieve better results, the use of more powerful optimization algorithms can be effective in achieving better results in software size estimation. This study presents an innovative approach to SCE that integrates the grey wolf optimization (GWO) algorithm to enhance the precision of ABE. The GWO algorithm, inspired by the hunting behavior and social hierarchy of grey wolves, is mathematically modeled and incorporated into the ABE approach. The key focus of this research is the optimization of the similarity function, a crucial component of the ABE, using both Euclidean and Manhattan distance measures. The article addresses the challenges in selecting an optimal similarity function and emphasizes the importance of proper feature weighting to improve estimation accuracy. The proposed GWO-based ABE method is rigorously evaluated on multiple software project datasets using cross-validation techniques. The performance of the GWO-based ABE is compared to other evolutionary algorithms based on widely accepted evaluation metrics. The results confirm that the integration of the GWO algorithm into ABE enhances estimation accuracy and model robustness. By optimizing feature weights in the similarity function, GWO-ABE effectively addresses key limitations of traditional analogy-based methods. The proposed approach demonstrates superior performance across multiple datasets, particularly under the Euclidean distance function, making it a reliable solution for software project cost estimation. Experimental evaluations show that GWO-ABE achieves notable improvements in key performance metrics, leading to reduced mean magnitude of relative error (MMRE), median magnitude of relative error (MdMRE), and higher percentage of prediction (PRED) compared to other ABE-customized methods. These findings highlight the role of metaheuristic optimization in improving software estimation techniques, contributing to more precise and efficient project planning and management.

# INTRODUCTION

Accurately estimating the cost of most software development is a pivotal concern at the outset of any software project[1]. The unique nature of the software products renders cost estimation a challenging endeavor. Precise cost estimation directly impacts the success or failure of software projects (*Dashti et al., 2022*). A survey conducted by the Project Management Institute (PMI) in 2017, indicated only 53% of projects have complied with the predicted budget. Furthermore, the results showed that 32% of the projects had exceeded the determined budget, and 16% had completely failed (*Langley, 2017*). According to recent reports, only 35% of projects are delivered on schedule, meet budgetary constraints, and align with predefined quality requirements, leaving 65% unable to achieve these objectives (*Dean, 2024*). Therefore, devising more efficient methods for SCE is of paramount importance for software specialists and researchers.

Reviewing the background reveals numerous methods and paradigms have been proposed for software cost estimation (SCE), some of which attempt to categorize these approaches. SCE relies on a diverse range of methods, which can be broadly categorized into algorithmic and non-algorithmic approaches. Algorithmic models, such as COCOMO, function point analysis (FPA), use case points (UCP), rely on mathematical formulas derived from historical project data, while non-algorithmic methods, including Planning Poker, and expert judgment, leverage human expertise and structured evaluation techniques to estimate effort (*Eduardo Carbonera, Farias & Bischoff, 2020*; *Jorgensen & Shepperd, 2006*; *Rashid et al., 2023*). Building upon these studies, our research specifically focuses on enhancing analogy-based estimation (ABE) through optimization techniques. Among these, the ABE stands out as one of the most popular SCE methods. Several studies have highlighted the effectiveness and superiority of ABE over traditional methods. For instance, *Gandomani et al. (2024)* noted that ABE's reliance on historical project data makes it a natural fit for decision-making in SCE. *AlMutlaq, Jawawi & Arbain (2021)* found that ABE often surpasses parametric models in terms of accuracy, particularly when a rich dataset of past projects is available. Inspired by instinctive decision-making of humans, this method was initially employed in SCE in 1997 and has since gained widespread adoption (*Naik & Nayak, 2017*). The accuracy of cost estimation in the ABE is dependent on four components similarity function, historical project sets, the number of the closest similarities, and the solution function. The method selects one or more previously developed projects similar to the project under development from the historical dataset based on the similarity function and predicts the cost required for executing the project based on similar projects.

In recent years, data-driven approaches, including machine learning and optimization-based models, have gained traction for their ability to improve estimation accuracy (*Sharma & Singh, 2017*; *Wen et al., 2012*). These techniques have played a crucial role in advancing software estimation methods. Researchers are actively exploring novel

---

[1] Portions of this text were previously published as part of a preprint (*Gandomani, Ansaripour & Dashti, 2024*).

methods in this domain owing to their substantial advancements in this field (*Alsaadi & Saeedi, 2022*; *Kaushik, Tayal & Yadav, 2022*; *Khan et al., 2022*). In particular, the ABE has been strengthened with the help of optimization algorithms. However, it seems that better results can be achieved using this method by using more powerful optimization algorithms.

Several comparative studies have analyzed the effectiveness of various machine learning techniques in software effort estimation, providing valuable insights into their accuracy and applicability. For instance, *Jayadi & Ahmad (2023)* compared multiple regression techniques, including linear regression, ridge regression, Least Absolute Shrinkage and Selection Operator (LASSO) regression, decision tree regression, and support vector regression, concluding that LASSO regression demonstrated the highest accuracy in feature selection and estimation performance. Similarly, *Rahman et al. (2024)* conducted an extensive empirical study evaluating K-nearest neighbor, support vector machine, Random Forest, logistic regression, and LASSO regression using benchmark datasets such as COCOMO, Albrecht, Desharnais, and Maxwell, highlighting the strengths of ensemble-based models in improving predictive accuracy. Furthermore, *Kumar & Singh (2020)* assessed linear regression, multi-layer perceptron (MLP), and Random Forest, revealing that linear regression consistently outperformed the other models in terms of estimation accuracy when implemented using the WEKA toolkit. Additional studies, such as those conducted by *Garg (2022)*, as well as *Mateen & Malik (2023)*, have further explored ensemble techniques, stochastic gradient boosting, and industry-specific comparisons, demonstrating how machine learning-based models can significantly enhance software cost estimation compared to traditional parametric methods.

Building on these insights, *Idri, Azzahra Amazal & Abran (2015)* proposed a classification specific to ABE methods with privileged attention to utilizing machine learning methods. Their categorization can be identified into two main categories: analogy-based techniques, which are used independently, and those combined with machine learning-based methods or other non-machine learning-based methods. The combination of models has to comply with a set of fusion rules recommended by *Wen et al. (2012)*. The analogy-based techniques may solely employ machine learning or use a combination of various techniques. Consequently, *Wen et al. (2012)* have identified eight pure machine learning methods. More recently, *Dashti & Gandomani (2022)* have introduced a comprehensive classification method, emphasizing hybrid methods and categorizing estimation methods into base and hybrid with a more in-depth investigation of hybrid methods.

Optimization techniques play a crucial role in improving the accuracy and reliability of software effort estimation models (*Ali & Gravino, 2021*; *Ansaripour & Gandomani, 2023*). Over the years, various optimization methods have been explored, among which computational intelligence (CI) and bio-inspired optimization approaches have gained significant attention. CI techniques, such as neural networks and fuzzy logic, leverage adaptive learning mechanisms to refine estimation models, while bio-inspired algorithms, including genetic algorithms (GA), particle swarm optimization (PSO), and ant colony optimization (ACO), draw inspiration from natural processes to optimize search and

decision-making. These algorithms have demonstrated effectiveness in feature selection, parameter tuning, and similarity function optimization within effort estimation models.

Among bio-inspired techniques, the grey wolf optimization (GWO) algorithm is inspired by the grey wolves (*Canis lupus*) and imitates the hunting mechanism and the leadership hierarchy of grey wolves in nature. Research indicated its desirable performance in solving classic engineering design problems and many real applications that outperforms many metaheuristic algorithms such as PSO, differential evolution (DE), GAs, ACO, evolutionary programming (EP), and other similar algorithms (*Mirjalili, Mirjalili & Lewis, 2014*). The GWO algorithm was selected for this study due to its unique theoretical and practical advantages. Theoretically, GWO provides a balanced mechanism for exploration and exploitation by replicating the structured social order and collaborative hunting behaviors characteristic of grey wolves (*Agarwal et al., 2018*). This mechanism helps avoid premature convergence, a common issue in other optimization algorithms. Practically, GWO has shown high adaptability and robustness in feature selection and parameter optimization tasks, making it particularly suitable for ABE. GWO has been widely applied in software cost estimation (SCE), demonstrating its effectiveness in optimizing effort prediction models. However, despite its success in SCE, no prior study has explored its integration with analogy-based estimation (ABE). This research aims to bridge this gap by leveraging GWO's optimization capabilities to enhance ABE's predictive accuracy, thus contributing a novel approach to software cost estimation (*Alsheikh & Munassar, 2023*; *Kassaymeh et al., 2024*; *Khan et al., 2021*; *Putri, Siahaan & Fatichah, 2021*).

The main motivation behind this research is the pressing requirement for more accurate SCE. As suggested by previous studies and industry reports, most software projects face budget problems with a vast majority exceeding the budget while others fail. This highlights the importance of developing better techniques for estimating software development costs. There has been a growing opportunity to improve SCE practices over the recent advances in machine learning and artificial intelligence. Optimization algorithms like the GWO have been proposed as a lever to the ABE method for improved cost estimate reliability. The goal of this study is to explore how GWO algorithm can be merged with ABE method so as to advance SCE field.

The remainder of the article is organized as follows: "ABE in SCE" overviews the ABE method, and "Grey Wolf Optimization (GWO)" reviews the related works. The GWO is summarized in "Related Work". The proposed method is presented in "Proposed Method: GWO-Based ABE", and the evaluation criteria, datasets, implementation details, and results are discussed in Sections "Evaluation and Result" and "Discussion". Finally, "Limitations" concludes the article.

## ABE IN SCE

ABE has become a mainstream choice in SCE due to its intuitive appeal and ability to adapt historical project data for predicting new project efforts (*Azzeh, Elsheikh & Alseid, 2017*; *Shahpar, Bardsiri & Bardsiri, 2021*). The method's straightforward process aligns with human decision-making, making it particularly valuable for practical applications. Recent

advancements have further enhanced ABE's capabilities. For instance, *Manchala & Bisi (2022)* proposed a hybrid model combining teaching-learning-based optimization with ABE, achieving improved prediction accuracy across multiple datasets. Similarly, *Jadhav et al. (2023)* introduced Omni-Ensemble Learning, which integrates multiple optimization algorithms to boost software effort estimation methods, showcasing the potential for combining ABE with modern machine learning approaches. These developments highlight ABE's adaptability and relevance in addressing the challenges of SCE across varied contexts. These findings confirm the robustness of ABE in diverse real-world scenarios, reinforcing its value in SCE research. ABE due to its intuitive nature and reliance on historical project data. It consists of four primary stages (*Abnane & Idri, 2018*):

- Gathering historical data: The first step involves collecting information on the previous projects and providing the historical datasets, which are available in different datasets. These datasets form the basis for identifying patterns and similarities between projects.
- Feature selection: In this stage, the key features of the new project are identified to align with the features available in the historical datasets. This ensures that meaningful comparisons can be made between projects.
- Similarity measurement: The similarity measure is calculated to determine how closely the new project resembles previous projects. The standard similarity functions used in this method include the Manhattan distance-based and Euclidean similarity. The methods for calculating these measures will be detailed in the following sections.
- Cost prediction: The cost of the new project is estimated based on the nearest neighbors identified through similarity measurement, utilizing the solution function for this purpose. Details of the prediction process, including the solution functions used, are provided later.

Since, in this research, the GWO is used for the similarity function to decrease the error rate as well as improve the performance of the ABE method, the functionality of the GWO similarity function is described in the following.

## Similarity function

The similarity function measures the similarity degree between two different projects and is a critical component in the ABE method. Since the similarity functions use different structures to measure the distance between projects, selecting various types may affect the projects selected as the nearest neighbors. Therefore, selecting the similarity function is of great importance. Among many different similarity functions, the Euclidean distance-based similarity and the Manhattan distance-based similarity are the most popular methods (*AlMutlaq, Jawawi & Arbain, 2023*; *Bardsiri et al., 2013*; *Benala & Mall, 2018*; *Dashti et al., 2022*; *Huang & Chiu, 2006*; *Khatibi Bardsiri & Hashemi, 2016*; *Nasr & Mohebbi, 2023*; *Shahpar, Bardsiri & Bardsiri, 2021*).

The similarity function is a fundamental component of the ABE method, as it determines the degree of resemblance between a new project and historical projects. This
function quantifies how closely the features of two projects align, thereby forming the basis for selecting the most similar historical projects, known as the nearest neighbors.

In its standard form, the similarity function can be expressed as follows (Eq. (1)):

$$sim(p.p') = f\left(Lsim\left(f_1.f_1'\right).\ Lsim\left(f_2.f_2'\right).\ldots.Lsim\left(f_n.f_n'\right)\right). \tag{1}$$

In this equation, the terms $p$ and $p'$ represent the prospective project and an existing project from the dataset, respectively (*Amazal, Idri & Abran, 2019*). The features of these projects are denoted by $f_i$, and $f_i'$, where n is the total number of features. The function $Lsim()$ computes the similarity for each pair of features, and the results are aggregated using $f()$, which synthesizes these individual similarity scores into a single comprehensive similarity value for the project pair.

The function operates in two main stages. First, for every feature of the projects, the similarity between the feature values of the current project and prior ones is computed using $Lsim(f_i, f_i')$. This process evaluates how closely each feature of the new project aligns with its counterpart in the historical dataset. The resulting similarity scores for all features are then combined using the $f()$, function, which generates a unified similarity score reflecting the overall resemblance between the projects (*Gautam & Singh, 2017*).

In our study, we extended this standard formulation by introducing two additional parameters, w and δ, alongside a feature weighting mechanism. The parameter w represents the weight assigned to each feature, taking values between 0 and 1 to account for the varying importance of features in the similarity calculation. Meanwhile, δ is a small constant added to the denominator to avoid division by zero and ensure numerical stability. These enhancements allow the similarity function to dynamically adjust to the characteristics of different datasets, making it more robust and effective in practice.

By incorporating these refinements, the similarity function not only ensures a more precise comparison of projects but also enhances the reliability of the ABE process, contributing to improved software effort estimation accuracy.

*Euclidean distance:* The Euclidean distance is a widely used similarity measure that calculates the straight-line distance between two projects in a multi-dimensional space. Mathematically, it is defined as Eq. (2):

$$sim(p.p') = \frac{1}{\left[\sqrt{\sum_{i=1}^{n} w_i Dis\left(f_i.f_i'\right) + \delta}\right]} \qquad \delta = 0.0001 \tag{2}$$

$$\begin{cases} \left(f_i - f_i'\right)^2 & \text{if features are numeric} \\ 1 & \text{if features are numeric and } f_i = f_i' \\ 0 & \text{if features are numeric and } f_i \neq f_i' \end{cases}$$

where p and p' are the two projects, $f_I$ and $f_i'$ represent the ith feature of projects p and p', respectively. $w_i \in [0\ .\ 1]$ is the weight of that feature. Furthermore, δ is a small constant to prevent the denominator from being zero, and n represents the number of total features.

The *Dis*() function represents the mathematical operation performed to calculate the distance or similarity. *Dis*() computes the squared differences between corresponding features of the projects, followed by summation and normalization.

*Manhattan distance:* This similarity is defined based on the Manhattan distance, which is the sum of total absolute value distances for each feature pair and is represented in Eq. (3).

$$\text{sim}(p.p') = \frac{1}{\left[\sum_{i=1}^{n} w_i \text{Dis}\left(f_i.f_i'\right) + \delta\right]} \qquad \delta = 0.0001 \tag{3}$$

$$\begin{cases} \left|f_i - f_i'\right| & \text{if features are numeric} \\ 1 & \text{if features are numeric and } f_i = f_i' \\ 0 & \text{if features are numeric and } f_i \neq f_i'. \end{cases}$$

As with the Euclidean distance, $p$, $p'$, $f_i$, $f_i'$, $w_i$, and $\delta$ retain the same definitions.

There are other types of similarity criteria in the background, such as maximum distance-based similarity, Minkowski distance-based similarity (*Angelis & Stamelos, 2000*), and rank mean similarity, which is the mean of ranking value of each feature of the project (*Walkerden & Jeffery, 1997*). Table 1 summarizes some similarity functions employed in previous studies.

Some background studies have compared the performance of different similarity functions; however, as Table 1 illustrates, the Euclidean and Manhattan distance-based similarity are the most popular ones due to the simple geometrical distance definition they provide for the distance between two points in a k-dimension Euclidean space.

Moreover, studies such as those (*Angelis & Stamelos, 2000*) concluded that the Euclidean, Manhattan, and maximum distance-based similarities provide almost the same results, probably affected by the dataset selection. This claim is also confirmed in a study performed by *Huang & Chiu (2006)*. In total, no solution for this problem determines when to prefer what similarity function so far. The present article employs the Euclidean and Manhattan similarity functions based on the results of Table 1.

According to the similarity function in relations Eqs. (1) and (2), it seems different features may have different significances for the similarity functions. For example, "function points (FPs)" are more important than the "programming language" in many cost models. Furthermore, numerous researchers state that there is a high potential for improving the precision of ABE by assigning proper weights to proper features. In this regard, multiple studies have determined the optimal weight for each feature (feature weighting). Some of the most important ones are introduced in the related studies section.

## K-nearest neighbors

The KNN algorithm is pivotal in analogy-based SCE. After calculating distances, KNN identifies the 'k' nearest projects (analogies) that are most similar to the unseen project. Choosing the appropriate "k" is vital for accurate results. The parameter 'k' denotes the count of closest neighbors considered in the estimation (*Ali & Gravino, 2021*). A small 'k' can lead to classifications that are highly sensitive to noise or outliers. In contrast, a large 'k'

**Table 1 A summary of the employed similarity functions.**

| Reference | Year | Euclidean distance | Manhattan distance | Maximum distance | Minkowski distance | Rank mean similarity |
|---|---|---|---|---|---|---|
| Shepperd & Schofield (1997) | 1997 | Yes | No | No | No | No |
| Walkerden & Jeffery (1997) | 1999 | No | No | No | No | Yes |
| Angelis & Stamelos (2000) | 2000 | Yes | Yes | Yes | No | No |
| Leung & Fan (2002) | 2002 | No | Yes | No | No | No |
| Mendes, Mosley & Counsell (2003) | 2003 | Yes | No | Yes | No | No |
| Molokken & Jorgensen (2003) | 2003 | Yes | No | No | No | No |
| Auer et al. (2006) | 2006 | Yes | No | No | No | No |
| Huang & Chiu (2006) | 2006 | Yes | No | No | No | No |
| Chiu & Huang (2007) | 2007 | Yes | Yes | No | Yes | No |
| Li et al. (2007) | 2007 | Yes | No | No | No | No |
| Mittas, Athanasiades & Angelis (2008) | 2008 | Yes | No | No | No | No |
| Li & Ruhe (2008) | 2008 | Yes | No | No | No | No |
| Keung, Kitchenham & Jeffery (2008) | 2008 | Yes | No | No | No | No |
| Li, Xie & Goh (2009b) | 2009 | Yes | Yes | No | No | No |
| Bardsiri et al. (2013) | 2013 | Yes | Yes | No | No | No |
| Benala & Mall (2018) | 2018 | Yes | Yes | No | No | No |
| Shah et al. (2020) | 2020 | Yes | Yes | No | No | No |
| Shahpar, Khatibi & Bardsiri (2021) | 2021 | Yes | Yes | No | No | No |
| Dashti et al. (2022) | 2022 | Yes | Yes | No | No | No |

may result in overly generalized decisions, missing important local data details. Therefore, it is crucial to assess the data's features and the specific project requirements to determine the optimal "k" value. This optimal value balances capturing significant patterns and minimizing the impact of noise.

In this study, we selected k = 3 after conducting preliminary experiments and reviewing similar studies in the field. Several studies have demonstrated that choosing k = 3 provides a balance between minimizing noise sensitivity and preserving the local structure of the data , leading to improved estimation accuracy (*Bardsiri et al., 2013*; *Benala & Mall, 2018*; *Dashti et al., 2022*). This value was chosen as it ensures that the algorithm captures meaningful patterns without overfitting or underfitting the model.

## Solution function

Solution function is a critical component in ABE methods, responsible for determining the predicted value (*e.g.*, effort) based on the characteristics of similar projects. It aggregates the outcomes of selected analogies to produce a single estimation value. Commonly used solution functions in ABE include closest analogy, mean, and median, as they balance the influence of outliers and ensure a robust estimate of software development effort (*Jorgensen & Shepperd, 2006*). Furthermore, addressing noise in

historical datasets and assigning appropriate weights to project features are essential for enhancing solution function reliability (*Madari & Niazi, 2019*). Research shows that choosing a suitable solution adaptation technique in ABE significantly impacts accuracy (*Phannachitta et al., 2017*). However, selecting an appropriate solution function relies on the data distribution and the particular demands of the estimation task (*Rashid et al., 2025*). In the following, different types of solution functions employed in software cost estimation are presented.

- Closest analogy method estimates cost by identifying the single most similar historical project and using its cost value as the estimate. This approach assumes that the most similar project provides the best reference for the new project.
- Mean method involves averaging the cost values of several similar projects to obtain an estimate. This technique helps to smooth out anomalies or variations in individual projects, providing a more balanced estimate.
- Median method calculates the cost by finding the middle value of the cost values from the selected similar projects. This approach reduces the impact of outliers and extreme values, offering a more robust estimate in the presence of skewed data.

## GREY WOLF OPTIMIZATION

The GWO was introduced by *Mirjalili, Mirjalili & Lewis (2014)* and is inspired by the hierarchical structure and hunting behavior of Grey wolves. It is initially discussed with a focus on the inspiration of the proposed method, then the mathematical model is provided.

- **Inspiration**

Grey wolves belongs to the Canidae family. Grey wolves are considered the apex predators, *i.e.*, they are situated at the top of the food chain and prefer to live in a pack. A pack usually includes 5–12 wolves. The precise social management of the pack is interesting; the wolves are divided into four subsections alpha, beta, omega, and delta. These subsections are, in fact, the components of a hierarchy, each of which has its duties. The main wolves, alphas, are comprised of a male and a female wolf, responsible for major decisions regarding hunting, sleeping location, wake-up time, *etc.*, (*Mech, 1999*). The second level includes the grey beta wolves; they are subordinate wolves, helping the alpha with decision-making or the other activities of the pack. The omega is the lowest ranked grey wolf, which has a more limited privilege compared to the other members. They are the last wolves allowed to eat. However, it should be noted that if the omega is eliminated, the pack confronts inner severe problems. If a wolf is not an alpha, beta, or omega, it is called a delta. The delta wolves are to be submissive to alphas and betas, yet dominant over omegas. These wolves protect the pack, guarantee the pack's immunity, and take care of weak, sick, and injured wolves. In addition to their social position, another interesting behavior of grey wolves is

their pack hunting. According to *Muro et al. (2011)* the main steps of grey wolf hunting are as follows:

1. Tracing, chasing, and approaching the prey.
2. Chasing, surrounding, and harassing the prey until it stops moving.
3. Attacking the prey.

- **Mathematic model and algorithm**

The alpha ($\alpha$) is considered the best solution in the mathematical modeling of the wolves' social hierarchy when designing GWO. Therefore, the second and third solutions are called beta ($\beta$) and delta ($\delta$), respectively. The rest of the candidate solutions are assumed as omega ($\omega$). In the GWO algorithm, the hunting is led by $\alpha$, $\beta$, and $\delta$. The $\omega$ wolves are submissive to these three wolves.

- **Prey surrounding equations (Eqs. (4) and (5))**

$$\vec{D} = \left| \vec{C}.\vec{X}_p(t) - \vec{X}(t) \right| \tag{4}$$

$$\vec{X}(t+1) = \vec{X}_p(t) - \vec{A}.\vec{D} \tag{5}$$

where t indicates the current iteration, $\vec{A}$ and $\vec{C}$ are the coefficient vectors, $\vec{X}_p$ is the position of the prey, and $\vec{X}$ shows the position vector of a grey wolf. The $\vec{A}$ and $\vec{C}$ vectors are calculated as Eqs. (6) and (7):

$$\vec{A} = 2\vec{a}.\vec{r}_1 - \vec{a} \tag{6}$$

$$\vec{C} = 2.\vec{r}_2 \tag{7}$$

where the components of $\vec{a}$ are decreased linearly from two to zero, and $\vec{r}_1$, $\vec{r}_2$ are random vectors in the interval [0, 1]. By setting the values of vectors $\vec{A}$ and $\vec{C}$, different positions around the best agent are achievable considering the current position.

- **Hunting**

For mathematical simulation of the hunting behavior of the Grey wolves, it is assumed that the alpha is the best-nominated solution, and beta and data have better information about the potential position of the prey. Therefore, we save the first three solutions as the best-captured solutions, and compel the rest of the searching agents—including the omega—to update their positions according to the positions of the best search agents. The Eqs. (8)–(10) are proposed in this regard:

$$D_\alpha = \left| \vec{C}_1.\vec{X}_\alpha - \vec{X} \right|. \ \vec{D}_\beta = \left| \vec{C}_2.\vec{X}_\beta - \vec{X} \right|. \ \vec{D}_\delta = \left| \vec{C}_3.\vec{X}_\delta - \vec{X} \right| \tag{8}$$

$$\vec{X}_1 = \vec{X}_\alpha - \vec{A}_1 . (\vec{D}_\alpha). \ \vec{X}_2 = \vec{X}_\beta - \vec{A}_2 . (\vec{D}_\beta). \ \vec{X}_3 = \vec{X}_\delta - \vec{A}_3 . (\vec{D}_\delta) \tag{9}$$

$$\vec{X}(t+1) = \frac{\vec{X}_1 + \vec{X}_2 + \vec{X}_3}{3} \tag{10}$$

- **Attacking the prey (exploitation)**

As mentioned above, the grey wolves finalize the hinting by attacking the prey, when it has stopped moving. To mathematically model the prey approaching, we decrease the amount of $\vec{a}$. Note that the oscillation range of $\vec{A}$ also decreases with $\vec{a}$. With the operands defined

```
Initialize the grey wolf population Xᵢ (i = 1, 2, … , n)
Initialize a, A, and C
Calculate the fitness of each search agent
Xₐ = the best search agent
X_β = the second best search agent
X_δ = the third best search agent
while (t < Max number of iterations)
    for each search agent
        Update the position of the current search agent by equation (9)
    end for
    Update a, A, and C
    Calculate the fitness of all search agents
    Update Xₐ, X_β, and X_δ
    t = t + 1
end while
return Xₐ
```

**Figure 1  Pseudocode of the grey wolf algorithm.**

so far, the GWO algorithm allows its search agents to update their positions according to the positions of alpha, beta, and delta, and attack the prey.

- **Search for prey (exploration)**

Grey wolves mostly search according to the position of the alpha, beta, and delta. They diverge to search for the prey and converge to attack the prey. For the mathematical modeling of divergence, we use $\vec{A}$ with random values greater than 1 or less than -1 to make the search agent diverge from the prey. This step emphasizes exploration, allowing the grey wolf algorithm to search globally. The other component of GWO helping the exploration is $\vec{C}$. This component helps the algorithm show a more random behavior during the optimization, supporting the exploration and preventing stuck in a local optimum. Vector $\vec{C}$ could be seen as the natural hurdle preventing the wolves from approaching the prey. Because the hurdles that exist in nature, appear on the path of the wolves and prevent their fast approach and readily attack the prey, which is applied by vector $\vec{C}$.

The pseudocode of the GWO is provided in Fig. 1.

## RELATED WORK

This section introduces the most important works related to ABE. *Li et al. (2007)* proposed a decision-based process model by modeling the existing effort estimation methods using similarity. The common decision-making problems are identified as a section of the model in different stages of the process, and a vast spectrum of alternative solutions are studied to solve these problems. Eventually, they provide a prototype of a process model. Then, in a different study, they proposed an adaptive method named AQUA, which overcomes the limitations of the previous methods—because they believed that the existing simulation-based methods are limited because of their inability to confront the uneven data anomalies (*Li, Al-Emran & Ruhe, 2007*). They proposed a hybrid of two ABE methods' ideas, including case-based reasoning and interactive filtering, which increases the prediction precision. The researchers also performed another study called AQUA+ in

2008. They conducted a qualitative analysis using the rough sequence analysis (RSA) to evaluate the features. The results of this study indicated that AQUA+ outperforms AQUA, and provides better results compared to the other ABE methods (*Li & Ruhe, 2008*).

Most researchers employ search-based software techniques using machine learning methods and collective intelligence to improve the performance of ABE methods (*Ahmed, Saliu & AlGhamdi, 2005*; *Azzeh, Neagu & Cowling, 2010*; *Galinina, Burceva & Parshutin, 2012*; *Jafari & Ziaaddini, 2016*). Most of these studies aim to enhance the precision of software development cost estimation. These studies could be divided into two groups feature weighting and feature selection. Since feature weighting is used in this current study, the focus in this section is more on the works related to this method.

The GA is the most prevalent optimization algorithm for calculating the weights of features. In 2006, *Huang & Chiu (2006)* concluded that the similarity metrics among feature pairs play a key role in analogy-based evaluation models. They investigated the influence of GA on the estimation precision of similarity metrics and used three similarity methods, *i.e.*, unequal weight method, linear weight method, and non-linear weight method. Their study results indicated the non-linear weight similarity outperforms other methods in terms of precision (*Huang & Chiu, 2006*). In 2009, *Li, Xie & Goh (2009b)* proposed a technique called the project selection technique for ABE (PSABE), in which a small subset of projects is selected, representing the whole main projects. The PSABE technique, then, combines the weight of the feature with a GA, called FWSPACE. Accordingly, they attempted to complete ABE (*Li, Xie & Goh, 2009b*). In 2015, *Kumari & Pushkar (2015)* employed the GA for selecting projects based on numerous criteria to improve the similarity estimation process; their method was suitable for reusing a previous project in defining a new project.

The PSO algorithm is another common method in SCE. In 2010, *Lin & Tzeng (2010)* initially used the Pearson product-moment correlation coefficient and one-way ANOVA to select multiple agents; then, they used the k-means clustering algorithm for clustering the software projects. After project clustering, they used PSO and optimized the parameters of the model.

*Bardsiri et al. (2013)* claimed that PSO computationally performs better than GA. In this regard, they used PSO for optimizing the weight in the similarity functions of the ABE model, which resulted in better identification of similar projects. They believed that the proposed method is sufficiently flexible to be used in different datasets. After that, *Liu et al. (2014)* attempted to use PSO to improve the estimation by minimizing errors in the training of datasets. They performed this task by optimizing non-orthogonal space distance (NoSD) according to the PSO algorithm. In 2016, *Azzeh et al. (2016)* stated in a study that three decision-making variables affect the successfulness of estimation, *i.e.*, the number of the nearest projects (k), the set of required optimal features for adaptation, and weight compliance. They employed PSO for identifying the optimal decision-making variable based on the optimization of multiple evaluation criteria in order to capture the exchange between different evaluation criteria.

*Bardsiri et al. (2012)* believed ABE and artificial neural networks (ANN) are the most popular methods used in cost estimation. Therefore, they proposed a hybrid

method of fuzzy clustering, ABE, and ANN, and utilized two big, real datasets to evaluate the performance of their proposed method; the results were eventually promising. They also studied the software services development effort estimation in another research, to create an efficient and reliable model by combining the ABE method and the DE algorithm (*Khatibi Bardsiri & Hashemi, 2016*). *Benala & Mall (2018)* studied the effectiveness of the DE algorithm for optimizing the feature weights of the ABE similarity functions using five mutation strategies. They simulated their studies over the experimental set of the PROMISE repository to capture the effectiveness of their proposed technique.

*Dashti et al. (2022)* investigated the learnable evolution model (LEM), and *Shah et al. (2020)* investigated the artificial bee colony (ABC) algorithm. They investigated guided ABE methods in two different studies in which the algorithms are combined with similarity functions and yield different weights for a more precise estimation; the most appropriate one in the ABE similarity function in the training stage is injected into the model, and then evaluated in the experiment stage.

Most studies in ABE have successfully utilized bio-inspired algorithms, such as PSO, ACO, GA, LEM, and ABC, to enhance feature weighting or improve similarity measures (*Ansaripour & Gandomani, 2023*). These methods have demonstrated considerable promise in addressing the challenges of SCE by leveraging natural phenomena to optimize complex problems. For example, *Nasr & Mohebbi (2023)* proposed a hybrid approach that combines PSO and GA to optimize feature weights in ABE. Their study demonstrated significant improvements in MMRE and PRED criteria on the Maxwell and Desharnais datasets. Similarly, *AlMutlaq, Jawawi & Arbain (2023)* introduced the firefly algorithm (FA) for feature weighting, achieving superior results compared to PSO and GA across multiple datasets. Moreover, *Gandomani et al. (2024)* extended the potential of ABE by integrating regression methods with weighted similarity functions, achieving higher accuracy and reliability in SCE.

Despite these advancements, each algorithm has its limitations, which can impact its performance in ABE. For instance, PSO is widely appreciated for its simplicity and ability to converge quickly to solutions. However, it often suffers from premature convergence, particularly in multi-dimensional spaces, where the algorithm may settle on local optima rather than the global optimum. This limitation can lead to suboptimal feature weighting in ABE, reducing the accuracy of cost estimation.

Similarly, ACO is an effective approach inspired by the behavior of ants in finding optimal paths. It excels in problems involving discrete search spaces and has been applied successfully to various optimization problems. However, ACO's high computational complexity can make it less suitable for real-time applications or large-scale datasets, such as those found in SCE. The time and resources required for the algorithm to iterate through multiple cycles can be prohibitive in practical scenarios.

GA, on the other hand, is a robust and versatile algorithm that simulates the process of natural selection. While GA is highly effective in exploring a wide search space, it often struggles with maintaining a balance between exploration and exploitation (*Vasuki, 2020*). This imbalance can result in slow convergence or the algorithm failing to fine-tune

solutions effectively, which can be a drawback in the context of ABE, where precision in feature weighting is crucial.

LEM presents a unique hybrid approach that combines evolutionary algorithms with machine learning techniques. While it has shown potential in certain domains, its applicability to ABE is limited by the need for extensive training data and computational resources. Additionally, the model's complexity may hinder its practical implementation, particularly when dealing with dynamic or incomplete datasets typical of SCE.

ABC, inspired by the foraging behavior of honey bees, is another notable bio-inspired algorithm. ABC algorithms are known for their ability to balance exploration and exploitation effectively, making them suitable for various optimization tasks. However, in the context of ABE, ABC may face scalability challenges due to the high communication overhead required to simulate bee colony dynamics. Moreover, optimizing its parameters to attain peak performance can be a resource-intensive and time-consuming process (*Nevena Rankovic, Ivanovic & Lazić, 2024*).

These limitations highlight the need for an alternative approach that not only addresses tackles the unique challenges of ABE while ensuring a balance between computational efficiency, convergence reliability, and adaptability to various datasets (*Srinadhraju, Mishra & Satapathy, 2024*). GWO was selected for this study due to its unique capability to balance exploration and exploitation, a critical factor in feature weighting for ABE. Unlike other methods, GWO mimics the social hierarchy and hunting behavior of grey wolves, enabling it to dynamically adjust the search process and avoid premature convergence. Furthermore, its simplicity and low computational overhead make it particularly suitable for large-scale datasets in software effort estimation. This choice aligns with the goal of enhancing ABE while addressing the limitations observed in previous methods.

The main objective of all related studies is to improve accuracy or reduce the error rate in SCE. Different studies attempted to use various methods to achieve this objective. As it seems using AI and machine learning algorithms has improved ABE, we use the grey wolf algorithm in the feature weighting method in this study, considering its main features and advantages.

# PROPOSED METHOD: GWO-BASED ABE

In this section, we propose a new ABE-customized method by integrating ABE and GWO. As mentioned before, feature weighting performs through the similarity function. Accordingly, we use GWO to define the weight of features in the stage of determining the similarity of projects. Generally, the proposed model is comprised of two parts. These parts are about model training and model testing. Each part is described in detail in the following.

## Model training

In this model, the effort (cost) feature in the datasets is considered the target feature or dependent feature, and the rest of the features are classified as independent features for software development cost estimation. In the training section, whole projects are initially divided into three different sets, *i.e.*, training (60%), validation (20%), and testing (20%).

This split is a widely adopted practice in machine learning and SCE studies as it ensures a balanced distribution of data for model training. The chosen 60/20/20 split aligns with recommendations from previous research, which suggest that this proportion offers a good balance between training and evaluation while maintaining the reliability of the results (*Bardsiri et al., 2013*; *Karimi & Gandomani, 2021*). In this setup, the training set is used to train the model, while the validation set plays a crucial role in hyperparameter tuning and model selection, ensuring that the trained model generalizes well before final evaluation. The test set remains completely unseen during training and validation and is solely used for final performance assessment. By maintaining this clear separation between training, validation, and testing, we ensure that the model's predictive performance is evaluated on entirely new data, reducing the risk of overfitting. The first two sets are used to train and optimize the model, and the third set is used to test the model. The training and testing projects are compared to the base projects for the appropriate weights to be found from the training projects and to evaluate the precision of the estimation model through the testing projects.

Initially, when the model is being trained, the weight of features could be set either equally (all zeros or ones, for example) or randomly. It should be noted that the random value must be chosen from interval [0, 1].

In this part, one project is selected as the target project in each iteration and undergoes Euclidean or Manhattan similarity functions. Every time the project goes through these functions, a series of optimized weights are generated in the interval [0, 1] and are assigned to the independent features. The optimized weights are generated using the grey wolf algorithm and are injected into the model. Then, the comparison operation is performed between the selected project and the training datasets for the KNN to be found.

When the most similar projects are found using the similarity measures defined in Eqs. (2) and (3), the amount of cost is calculated through the solution function—which can be mean or median—for the selected project. In fact, the most similar projects refer to the projects in the historical dataset that exhibit the highest similarity to the new project based on their features. The captured cost is evaluated using our intended evaluation criterion, *i.e.*, MRE. This trend is repeated until there is no project to be selected in the training datasets. The architecture of the training part is shown in Fig. 2.

## Model testing

In this part, test datasets are used to determine the precision performance of the model trained in the previous part. This part is mainly similar to the previous part. The only difference is that this part uses the optimized weights generated in the model training part, and the grey wolf algorithm is not executed again.

In this part, a project is selected from test datasets and sent to the similarity functions; then, the generated optimal weights in the model testing part are put alongside each feature as coefficients and, like the previous method, the number of KNN and the solution function is executed as well. Figure 3 shows the architecture of the testing part.

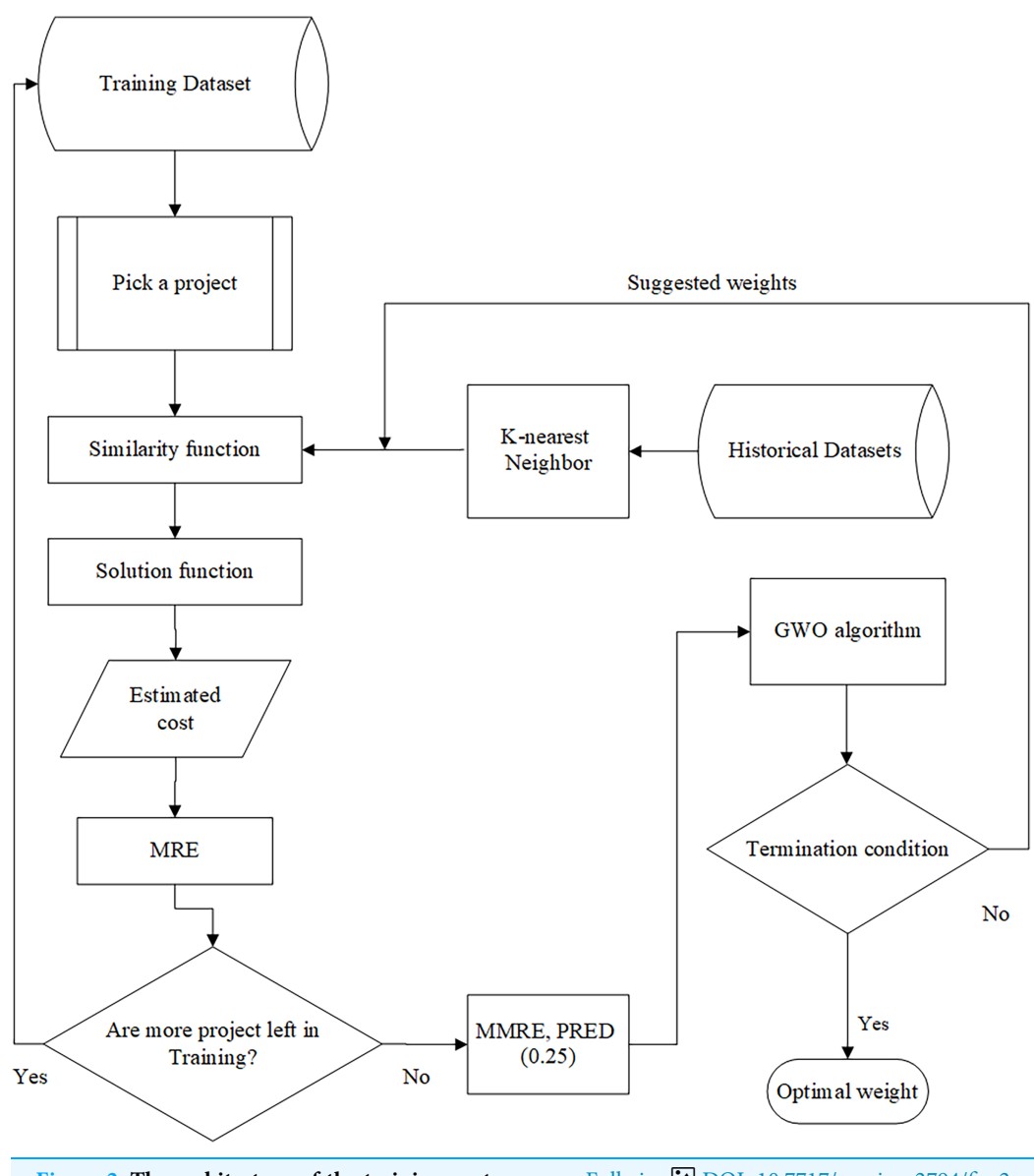

**Figure 2** **The architecture of the training part.**

## EVALUATION AND RESULTS

### Evaluation criteria

Evaluation criteria are essential for the experimental validation of cost estimation methods. In this study, we evaluated the performance of the proposed model using MMRE and PRED(q), which are widely adopted in ABE research. These metrics were selected because they specifically measure the relative estimation error, making them more suitable for assessing the accuracy of effort estimation models (*Idri, Azzahra Amazal & Abran, 2015*; *Jorgensen & Shepperd, 2006*). While absolute error-based metrics such as mean absolute error (MAE) and mean squared error (MSE) are commonly used in regression-based models, they do not account for the proportionality of the error in relation to the actual

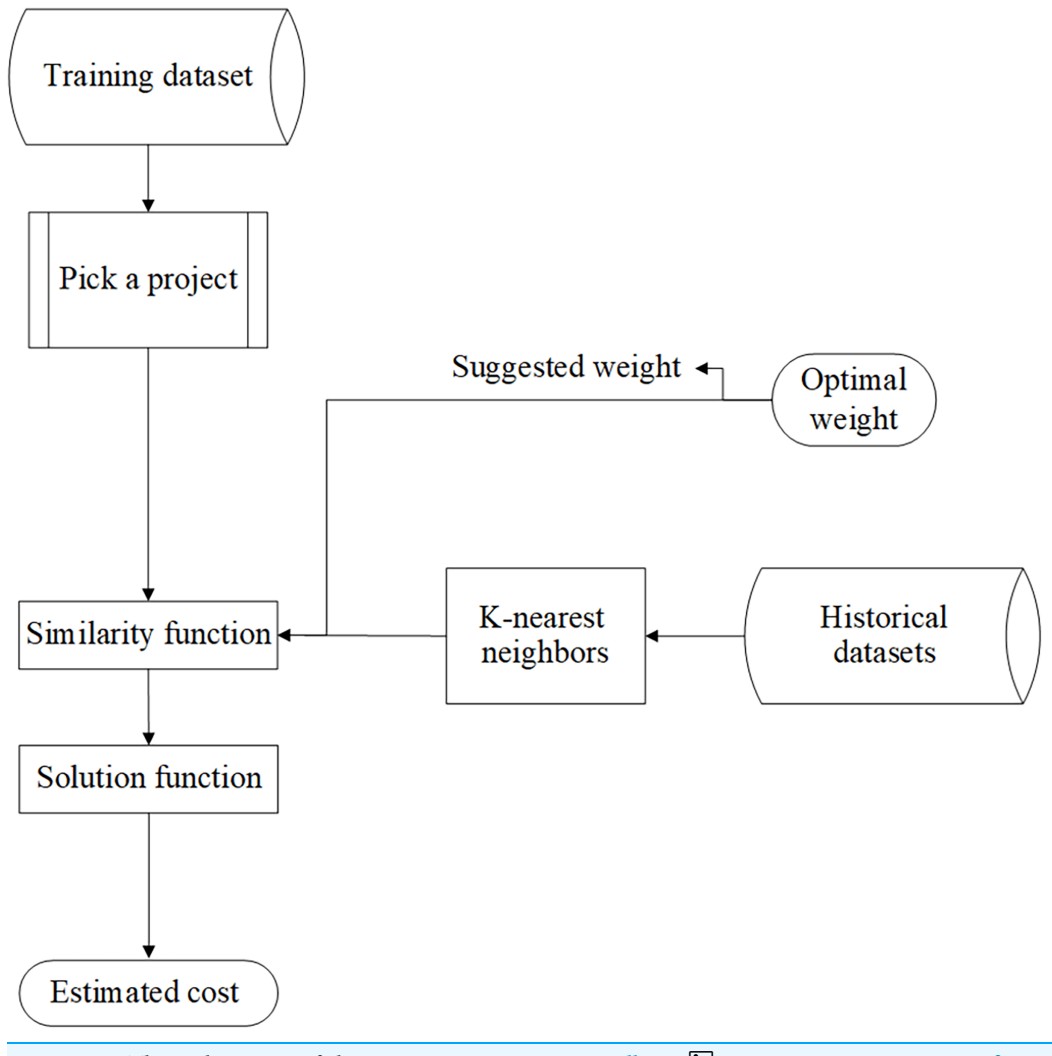

**Figure 3 The architecture of the testing part.**

effort values, which is crucial in ABE studies. Additionally, R² is more applicable in models where the goal is to measure variance explained by independent variables, whereas in ABE, the focus is on minimizing relative estimation errors to improve prediction accuracy. In the following, each evaluation criterion is introduced separately.

*The magnitude of relative error (MREi):* Error criterion in the class of relative error-based criteria, known as MREi, is defined as Eq. (11):

$$MRE_i = \left| \frac{C_i - \hat{C}_i}{C_i} \right| \tag{11}$$

where $C_I$ indicates the actual cost of the ith project, and $\hat{C}_I$ represents the estimated cost for the ith project.

*The mean magnitude of relative error (MMRE):* MMRE is defined as Eq. (12):

$$\text{MMRE} = \frac{1}{n} \times \sum_{i=1}^{n} \left| \frac{C_i - \hat{C}_i}{C_i} \right| = \text{Mean (MRE)} \tag{12}$$

where n indicates the number of projects to be estimated, $C_I$ indicates the actual cost of the ith project, and $\hat{C}_I$ represents the estimated cost for the ith project. Low amounts of MMRE indicate a low level of estimation error.

*Prediction at level q (PRED(q)):* PRED is the percentage of the predictions lying in a specific percentage of the actual cost (Eq. (13)):

$$\text{PRED(q)} = \frac{1}{n} \times \sum_{i=1}^{n} \delta(\text{MRE}_i - q) \tag{13}$$

$$\delta(x) = \begin{cases} 1 & x \leq 0 \\ 0 & x > 0 \end{cases}$$

where q is a predefined threshold. PRED(q) calculates the percentage of predictions whose MRE values are equal to or less than q. Most of the articles set q to 0.25 or 0.30. We set q to 0.25 because the comparison articles set the same value for q (*Ahadi & Jafarian, 2016*; *Beiranvand & Zare Chahooki, 2023*; *ul Hassan et al., 2022*).

*The median magnitude of relative error (MdMRE):* The MdMRE is the median of the MREs, which is shown in Eq. (14) and is calculated using the median function.

$$\text{MdMRE} = \text{median(MRE)}. \tag{14}$$

## Datasets

We used four different, freely available datasets in this study, which are vastly utilized in the research community of software effort estimation. We believe utilizing different datasets may help provide a more transparent evaluation. Research has shown that leveraging multiple datasets in software effort estimation can provide more robust and generalizable outcomes (*Rahman, Goncalves & Sarwar, 2023*). The utilized datasets in this study include Desharnais, Albrecht, China, and Maxwell. Table 2 provides the details of these datasets, including the number of features and their related information.

The Desharnais dataset contains data from 81 real-world software projects, making it one of the most commonly used datasets in this domain. It provides essential attributes such as the actual effort required for project completion (measured in person-months), the size of the development team, the function point's number (FP) as a measure of software functionality, and the total development time in months (*Karimi, Gandomani & Mosleh, 2023*). These attributes are particularly relevant for estimating software development efforts. The Albrecht dataset, introduced by Albrecht and Gaffney in 1983, originates from IBM software projects. It has been pivotal for FPA and software project estimation, exploring the relationship between software size and the effort required for development. The dataset includes information about the number of function points, source lines of code (SLOC), and actual effort in person-hours, which are key metrics for assessing software

**Table 2 The utilized datasets.**

| Dataset | No. of projects | No. of features |
|---|---|---|
| Albrecht | 24 | 7 |
| Desharnais | 77 | 11 |
| Maxwell | 62 | 27 |
| China | 499 | 19 |

project performance. The China dataset includes information on 499 software projects executed by Chinese companies (*Corazza et al., 2013*). It includes 19 distinct attributes that focus primarily on functional components such as input, output, query, file, and interface. These components are integral to calculating function points, a critical metric for estimating both software size and development effort. The substantial size and extensive features of this dataset render it highly valuable for assessing the generalizability of SCE models (*Wen et al., 2012*). The Maxwell dataset comprises 62 industrial software projects, predominantly developed in Finland by major commercial banks (*Ali & Gravino, 2021*). This dataset provides insights into real-world software development processes and includes attributes such as project size, measured in terms of lines of code (LoC), qualitative metrics for project complexity, and the total effort required for each project (measured in person-hours). Its focus on industrial applications enhances its relevance to real-world scenarios. All datasets and developed codes used in this study are available in the Supplemental Files to ensure the reproducibility of our work.

These datasets indicate an exciting set of software projects, including a unique software company or different companies' data, illustrating various application domains and projects' features.

## Experiment design

In SCE problems, data preprocessing is necessary before injecting it into the model. Otherwise, the training quality will decrease dramatically. For this, we use the min-max equation for the values of the independent features to lie in the interval [0, 1], so-called normalization (Eq. (15)). Normalization makes all features have a similar impact on the dependent feature.

$$X' = \frac{X - X_{min}}{X_{max} - X_{min}} \tag{15}$$

where $X$ represents the original value, $X_{min}$ and $X_{max}$ denote the minimum and maximum values of the feature, respectively, and $X'$ and is the normalized value (*Krishnan et al., 2021*). This approach has been extensively employed in research on SCE (*Wani, Giri & Bashir, 2019*).

In the next step of the model training, we should divide the datasets into training and testing datasets. For this, we use the cross-validation method. Here, we employ a three-fold cross-validation technique, which is widely used in SCE studies (*Amazal, Idri & Abran, 2014*; *Huang & Chiu, 2006*; *Li, Xie & Goh, 2009a*; *Rahman et al., 2023*). Datasets in this

**Table 3  Three-fold cross-validation.**

|  | Dataset | | |
|---|---|---|---|
|  | **Base dataset (Set1)** | **Training dataset (Set2)** | **Testing dataset (Set3)** |
| Fold1 | set1 | set2 | set3 |
|  | set1 | set3 | set2 |
| Fold2 | set2 | set1 | set3 |
|  | set2 | set3 | set1 |
| Fold3 | set3 | set2 | set1 |
|  | set3 | set1 | set2 |

method are threefold base, training, and testing datasets, which are selected randomly from the intended dataset. In this process, the data is partitioned into three groups: two groups are utilized as training datasets, while the remaining group is designated as the testing dataset. This division is repeated three times to ensure that each group serves as the testing dataset once (*Guo et al., 2023*). Table 3 shows the three-fold cross-validation.

About the cross-validation, it should be noted that the number of projects in each set is the same. The performance is measured for two separate sequences in each step, whose mean is considered as the resulting step. The final results are determined by averaging the mean values across the three steps, ensuring consistency and reliability in the performance evaluation.

We use the Manhattan and Euclidean similarity functions in this study. We set k = 3 for the number of nearest neighbors because most articles have set this value for k (*Alsaadi & Saeedi, 2022*; *Benala & Mall, 2018*; *Dashti et al., 2022*). We use the mean and median for the solution function. To design the experiment, we systematically evaluated different combinations of the control parameters—similarity function, k value, and solution function—to identify the optimal configuration. Finally, we select the fittest resulting model to be compared to other algorithms.

In the GWO algorithm, the encoding scheme represents the feature weights in ABE as a vector comprising continuous values bounded within the range [0, 1] (*Shahpar, Bardsiri & Bardsiri, 2021*). Each element of the vector corresponds to the weight assigned to a particular feature in the similarity function, ensuring consistency with the normalized data. The fitness function evaluates the quality of each candidate solution based on its ability to minimize the mean magnitude of relative error (MMRE). The fitness value for each wolf is calculated as Eq. (16):

$$Fitness = \frac{1}{n}\sum_{i=1}^{n}\left|\frac{Actual\ Effort_i - Estimated\ Effort_i}{Actual\ Effort_i}\right| \tag{16}$$

where *Actual Effort* is the observed effort for a project, *Estimated Effort* is the predicted effort, and n is the total number of projects in the training set.

For GWO, the population size was set to 8 as it produced the best results in our preliminary experiments. This value was chosen based on empirical testing to balance exploration and exploitation while ensuring computational efficiency. The number of iterations was set to 200, based on our preliminary experiments, which indicated that this value provides a balance between convergence speed and computational cost. A higher number of iterations showed diminishing improvements in accuracy, while lower iterations led to insufficient convergence. This approach ensures that the optimization process runs efficiently without unnecessary computational overhead. The convergence parameters (a, A, and C) were determined following the approach outlined in *Mirjalili, Mirjalili & Lewis (2014)*. The parameter a linearly decreases from 2 to 0 over iterations, allowing the algorithm to transition from exploration to exploitation. The coefficient A and C is dynamically adjusted using the Eqs. (6) and (7). These adjustments help the algorithm maintain an effective balance between global and local search, preventing premature convergence to local optima.

A standardized experimental setup was implemented to ensure the results' reliability. All algorithms were tested on the same datasets, under identical preprocessing and evaluation conditions. Each algorithm was executed 30 times to account for the stochastic nature of metaheuristic optimization, and the reported results represent the mean performance over these runs to ensure statistical reliability. For a fair comparison, we adopted the default parameter settings for baseline methods from their original research articles, as these configurations have been optimized and widely validated in prior studies. This ensures that each algorithm operates under its best-recommended settings without arbitrary modifications that could introduce bias. Furthermore, all experiments were conducted on the same hardware and software environment to eliminate any discrepancies due to computational resources. The datasets used in this study remained unchanged across all experiments, ensuring that each algorithm was evaluated on identical data partitions. Table 4 summarizes the key parameters and their values used in this study to provide a clear overview of the experimental settings.

## Execution time analysis

To provide a fair comparison of computational efficiency, we measured the execution times of all AI algorithms under identical hardware and software conditions. Table 5 presents these times, demonstrating that GWO-ABE maintains a competitive efficiency performance.

The results indicate that while traditional ABE is the fastest due to its simplicity, evolutionary algorithms such as GA, PSO, and ABC require longer execution times due to their iterative optimization processes. GWO effectively balances accuracy and computational efficiency, outperforming several metaheuristic algorithms while maintaining a competitive execution speed.

This observation is consistent with previous research on GWO, which demonstrated superior exploration and exploitation balance, better convergence speed, and robustness in

**Table 4 Summary of experimental parameters and settings.**

| Parameter | Description | Value used |
|---|---|---|
| Similarity function | Distance metric for analogy comparison | Euclidean, Manhattan |
| k (nearest neighbors) | Number of nearest neighbors in ABE | 3 |
| Optimization algorithm | The metaheuristic used | GWO |
| Population size | Number of search agents in GWO | 8 |
| Number of iterations | Iterations in optimization process | 200 |
| Solution function | Function used for cost estimation | Mean, Median |
| Evaluation metrics | Metrics used to assess model performance | MMRE, MdMRE, PRED(0.25) |
| Normalization method | Data scaling technique | Min-Max Normalization |
| Cross-validation | Validation technique used | 3-fold cross-validation |
| Baseline methods | Parameter settings for GA, PSO, and ABC | Adopted from original research articles |

**Table 5 Execution time comparison of artificial intelligence (AI) algorithms.**

| Algorithm | Execution time (seconds) |
|---|---|
| ABE (Baseline) | 12.3 |
| GA-ABE | 18.7 |
| PSO-ABE | 15.4 |
| ABC-ABE | 20.1 |
| GWO-ABE | 14.6 |

solving optimization problems compared to PSO and DE (*Mirjalili, Mirjalili & Lewis, 2014*; *Shial, Sahoo & Panigrahi, 2023*). GWO's ability to dynamically balance exploration and exploitation through adaptive parameter tuning allows it to avoid local optima more effectively, leading to higher accuracy with relatively low computational cost compared to other evolutionary techniques.

## Efficiency comparison

For comparing different utilized algorithms in this field, it should be considered that datasets, evaluation criteria, and sampling distances are selected the same, so the results are reliable. This research attempts to compare the performance of the proposed model with other methods. One consideration is that the evaluation criteria of the estimator functions are the same to perform a sound comparison. As mentioned before, the value of k is set to three. The performance of the model is initially tested with the base model ABE and then tested with evolutionary algorithms used in this field, including PSO-ABE, GE-ABE, DABE, ABC-ABE, and LEMABE. These methods were selected as they are frequently cited in the literature and have been extensively analyzed in the related work section. All mentioned estimation models are trained using historical datasets, ensuring consistency in evaluation, while the algorithm parameters are set automatically based on their recommended configurations from previous studies.

The required experiments were conducted on a PC equipped with a Ryzen 7900 CPU, an Nvidia RX 580 8GB GPU, and 16GB of RAM. Additionally, the model was

**Table 6 Project cost estimation results in Desharnais dataset compared to other ABE-customized methods.**

| Similarity function | Solution function | Response function | ABE | GA-ABE | PSO-ABE | DABE | LEMABE | ABC-ABE | GWO-ABE |
|---|---|---|---|---|---|---|---|---|---|
| Euclidean | Mean | MMRE | 0.3190 | 0.36508 | 0.46717 | 0.31061 | 0.34079 | 0.48797 | 0.31265 |
| | | MdMRE | 0.2222 | 0.33333 | 0.52778 | 0.36111 | 0.38889 | 0.41667 | 0.50 |
| | | PRED | 0.1818 | 0.26551 | 0.27273 | 0.3254 | 0.42857 | 0.37302 | 0.5671 |
| Euclidean | Median | MMRE | 0.53992 | 0.53247 | 0.31109 | 0.40356 | 0.32071 | 0.80147 | 0.41438 |
| | | MdMRE | 0.61111 | 0.66667 | 0.16667 | 0.38889 | 0.33333 | 0.66667 | 0.66667 |
| | | PRED | 0.23232 | 0.29437 | 0.40332 | 0.33983 | 0.53576 | 0.41775 | 0.57319 |
| Manhattan | Mean | MMRE | 0.80438 | 0.52862 | 0.48148 | 0.58249 | 0.39899 | 0.65067 | 0.49391 |
| | | MdMRE | 0.97778 | 0.44444 | 0.59259 | 0.77778 | 0.52778 | 0.66667 | 0.12121 |
| | | PRED | 0.09697 | 0.19048 | 0.18182 | 0.24242 | 0.27273 | 0.045455 | 0.61111 |
| Manhattan | Median | MMRE | 0.34909 | 0.39526 | 0.49222 | 0.84596 | 0.29293 | 0.4326 | 0.3634 |
| | | MdMRE | 0.33333 | 0.52778 | 0.24669 | 0.83333 | 0.33333 | 0.3142 | 0.38889 |
| | | PRED | 0.32756 | 0.2619 | 0.42279 | 0.33333 | 0.60606 | 0.6470 | 0.59524 |

**Table 7 Project cost estimation results in Maxwell dataset compared to other ABE-customized methods.**

| Similarity Function | Solution Function | Response function | ABE | GA-ABE | PSO-ABE | DABE | LEMABE | ABC-ABE | GWO-ABE |
|---|---|---|---|---|---|---|---|---|---|
| Euclidean | Mean | MMRE | 0.70685 | 0.7585 | 0.95527 | 0.56948 | 0.45036 | 0.29997 | 0.53684 |
| | | MdMRE | 0.37954 | 0.38117 | 0.58796 | 0.24877 | 0.25833 | 0.28571 | 0.2254 |
| | | PRED | 0.35294 | 0.22181 | 0.40441 | 0.40196 | 0.56863 | 0.53186 | 0.71569 |
| Euclidean | Median | MMRE | 0.40418 | 1.2613 | 0.41049 | 0.25915 | 0.5628 | 0.61111 | 0.38157 |
| | | MdMRE | 0.34722 | 0.9375 | 0.34921 | 0.17778 | 0.23056 | 0.22222 | 0.13889 |
| | | PRED | 0.14216 | 0.42525 | 0.47549 | 0.56863 | 0.71936 | 0.78064 | 0.82353 |
| Manhattan | Mean | MMRE | 0.34161 | 0.34566 | 0.4822 | 1.2498 | 0.38598 | 0.4837 | 0.4231 |
| | | MdMRE | 0.20106 | 0.17593 | 0.16138 | 0.87963 | 0.22906 | 0.28241 | 0.29762 |
| | | PRED | 0.68627 | 0.6973 | 0.70588 | 0.37255 | 0.65441 | 0.40441 | 0.42647 |
| Manhattan | Median | MMRE | 0.45532 | 0.55905 | 0.66391 | 1.0408 | 0.55349 | 0.28048 | 0.51626 |
| | | MdMRE | 0.42619 | 0.30952 | 0.28373 | 0.83333 | 0.26111 | 0.21429 | 0.19511 |
| | | PRED | 0.23529 | 0.38358 | 0.36275 | 0.40564 | 0.68627 | 0.68137 | 0.6973 |

implemented using MATLAB 2022. Tables 6–9 display the comparative results across various datasets.

As can be seen from the above tables, the amount of MMRE and MdMRE is lower in the GWO-ABE algorithm compared to the other algorithms. For a more transparent indication, in Figs. 4 to 7, we show two methods according to the Euclidean similarity of different sets. Next, we evaluate these two with the Manhattan similarity, whose results are shown in Figs. 8 to 11.

In Figs. 8 to 11, it can be seen that the proposed algorithm has managed to yield better results compared to the other algorithms. It should be noted in the charts that the proposed algorithm achieves better results in the Euclidean distance compared to the Manhattan distance. As mentioned before, the PRED (0.25) is another evaluation criterion, which is investigated. For the resulting information from this criterion to be represented

**Table 8 Project cost estimation results in China dataset compared to other ABE-customized methods.**

| Similarity function | Solution function | Response function | ABE | GA-ABE | PSO-ABE | DABE | LEMABE | ABC-ABE | GWO-ABE |
|---|---|---|---|---|---|---|---|---|---|
| Euclidean | Mean | MMRE | 1.4796 | 1.1209 | 0.56695 | 1.0179 | 0.93469 | 0.71703 | 0.6083 |
|  |  | MdMRE | 0.8744 | 0.6553 | 0.33986 | 0.1479 | 0.58273 | 0.38016 | 0.3182 |
|  |  | PRED | 0.087719 | 0.1328 | 0.49875 | 0.5720 | 0.50376 | 0.5213 | 0.5288 |
| Euclidean | Median | MMRE | 1.2353 | 0.69445 | 0.71826 | 0.99567 | 0.6948 | 0.88975 | 0.55912 |
|  |  | MdMRE | 0.85648 | 0.50387 | 0.47821 | 0.53297 | 0.3651 | 0.44963 | 0.26709 |
|  |  | PRED | 0.16792 | 0.16792 | 0.20802 | 0.17544 | 0.5088 | 0.51378 | 0.54637 |
| Manhattan | Mean | MMRE | 1.8934 | 1.4125 | 0.68333 | 0.97565 | 0.78542 | 0.71158 | 0.6862 |
|  |  | MdMRE | 1.2063 | 0.99499 | 0.39841 | 0.55 | 0.48413 | 0.43333 | 0.5556 |
|  |  | PRED | 0.085213 | 0.15539 | 0.18045 | 0.18546 | 0.18546 | 0.19048 | 0.2130 |
| Manhattan | Median | MMRE | 1.2674 | 0.6627 | 1.0649 | 1.726 | 0.62217 | 0.46034 | 0.9903 |
|  |  | MdMRE | 0.69048 | 0.39099 | 0.61905 | 0.92976 | 0.28376 | 0.31339 | 0.5259 |
|  |  | PRED | 0.16291 | 0.19048 | 0.20802 | 0.4386 | 0.50125 | 0.49123 | 0.5038 |

**Table 9 Project cost estimation results in Albrecht dataset compared to other ABE-customized methods.**

| Similarity function | Solution function | Response function | ABE | GA-ABE | PSO-ABE | DABE | LEMABE | ABC-ABE | GWO-ABE |
|---|---|---|---|---|---|---|---|---|---|
| Euclidean | Mean | MMRE | 0.73303 | 0.43954 | 0.42897 | 0.38054 | 0.32836 | 0.42472 | 0.36446 |
|  |  | MdMRE | 0.77814 | 0.41059 | 0.42546 | 0.38161 | 0.33895 | 0.55556 | 0.3795 |
|  |  | PRED | 0.11111 | 0.15873 | 0.38889 | 0.48413 | 0.49206 | 0.50361 | 0.55556 |
| Euclidean | Median | MMRE | 0.49931 | 0.6102 | 0.53359 | 0.42549 | 0.71868 | 0.40343 | 0.15404 |
|  |  | MdMRE | 0.50458 | 0.65879 | 0.60318 | 0.44456 | 0.71489 | 0.41893 | 0.083333 |
|  |  | PRED | 0.095238 | 0.10317 | 0.10317 | 0.15079 | 0.20635 | 0.30159 | 0.72727 |
| Manhattan | Mean | MMRE | 0.48583 | 0.47416 | 0.42881 | 0.80701 | 0.41023 | 0.87233 | 0.38648 |
|  |  | MdMRE | 0.47204 | 0.4565 | 0.45207 | 0.70844 | 0.41615 | 0.54292 | 0.44444 |
|  |  | PRED | 0.0125 | 0.047619 | 0.33333 | 0.20635 | 0.44444 | 0.38095 | 0.55195 |
| Manhattan | Median | MMRE | 0.47482 | 0.47416 | 0.80701 | 0.42881 | 0.41023 | 0.87233 | 0.38648 |
|  |  | MdMRE | 0.44125 | 0.4565 | 0.70844 | 0.45207 | 0.41615 | 0.54292 | 0.44444 |
|  |  | PRED | 0.0259 | 0.047619 | 0.20635 | 0.33333 | 0.44444 | 0.38095 | 0.55195 |

with more clarity, we express it using the Euclidean and Manhattan distances as well as mean and mean solution functions separately for different datasets in Figs. 12 to 15.

As can be seen from Figs. 12 to 15, the PRED (0.25) criterion in the proposed algorithm has managed to yield more desirable results.

## DISCUSSION

This section compares and discusses the results of different evolutionary algorithms (ABE, GA-ABE, PSO-ABE, DABE, LEMABE, ABC-ABE, GWO) for project cost estimation using the different datasets, specifically focusing on various metrics (MMRE, MdMRE, PRED) under different similarity (Euclidean and Manhattan) and solution functions (mean and median).

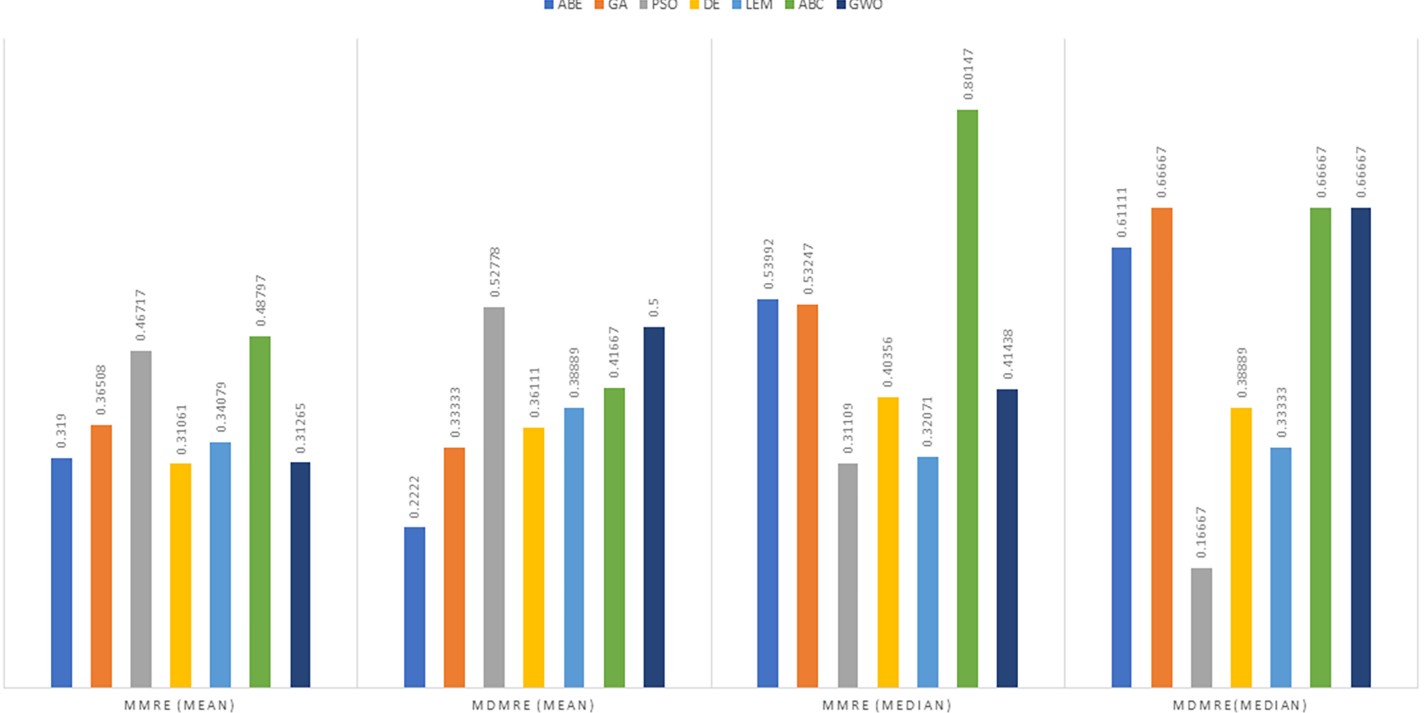

**Figure 4 Evaluation representation of MMRE and MdMRE criteria in the Desharnais datasets using the Euclidean similarity function.**

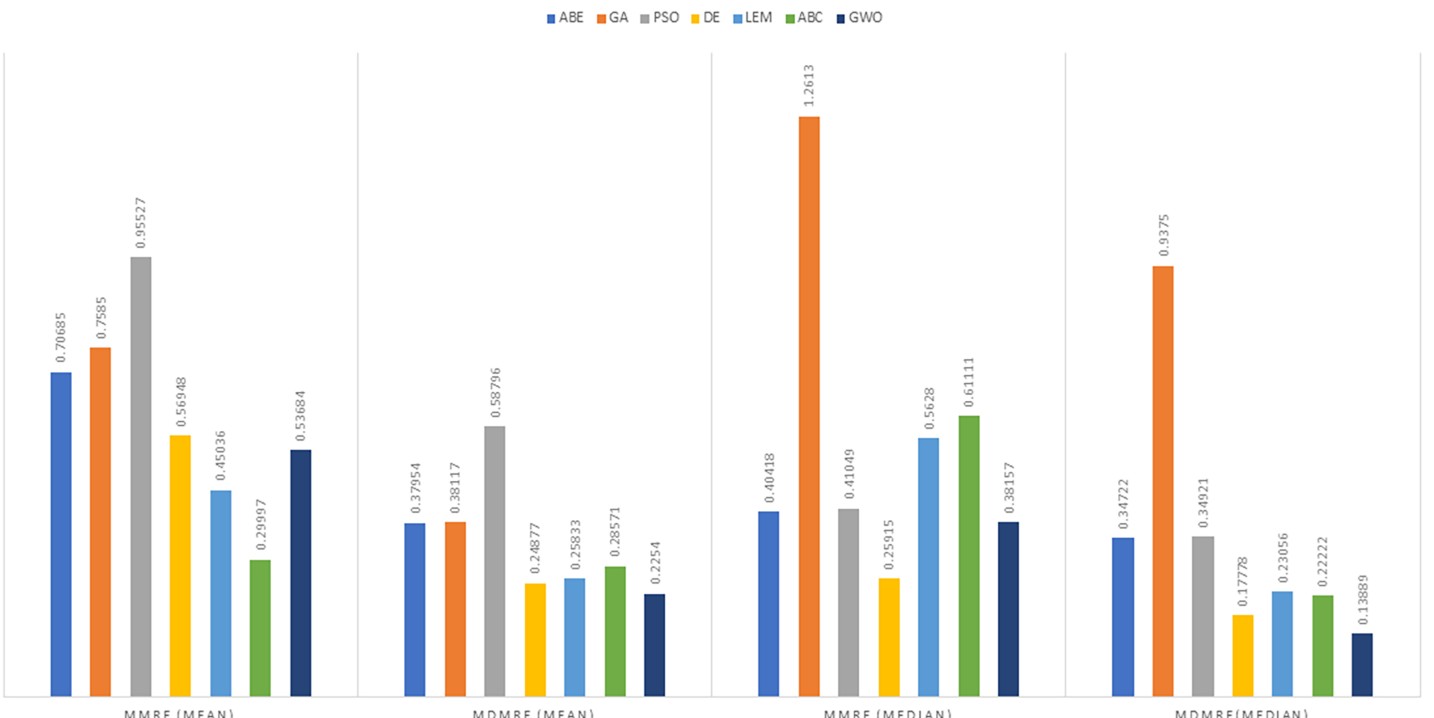

**Figure 5 Evaluation representation of MMRE and MdMRE criteria in the Maxwell datasets using the Euclidean similarity function.**

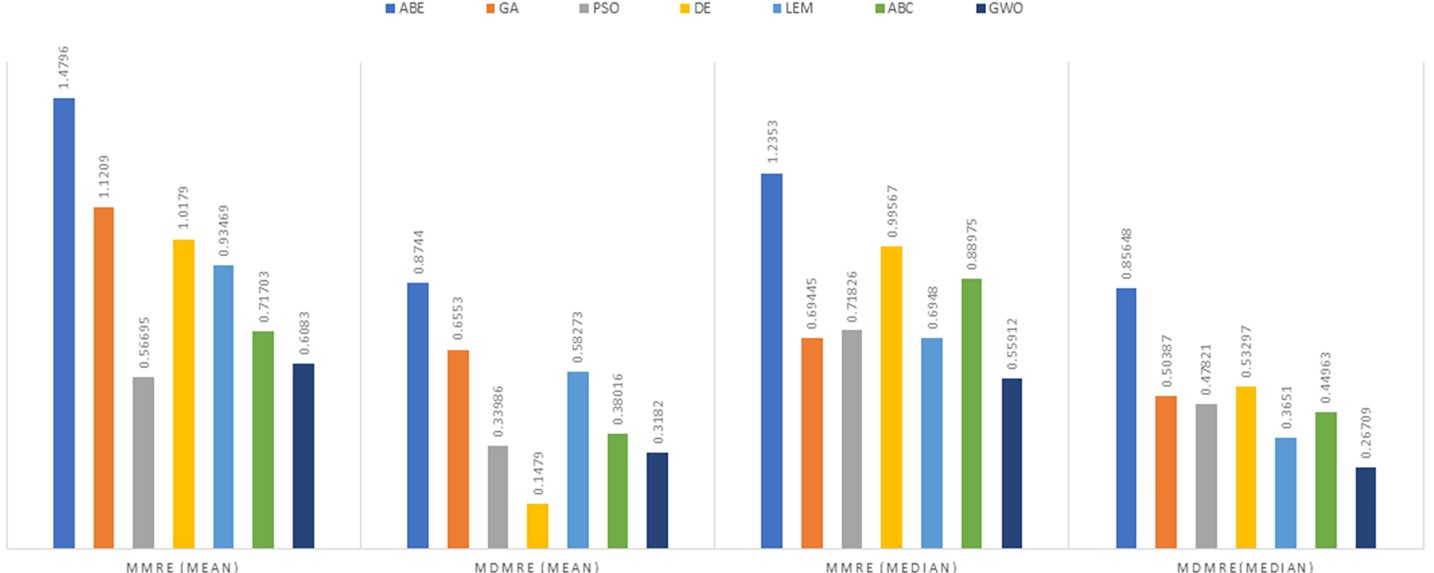

**Figure 6** Evaluation representation of MMRE and MdMRE criteria in the China datasets using the Euclidean similarity function.

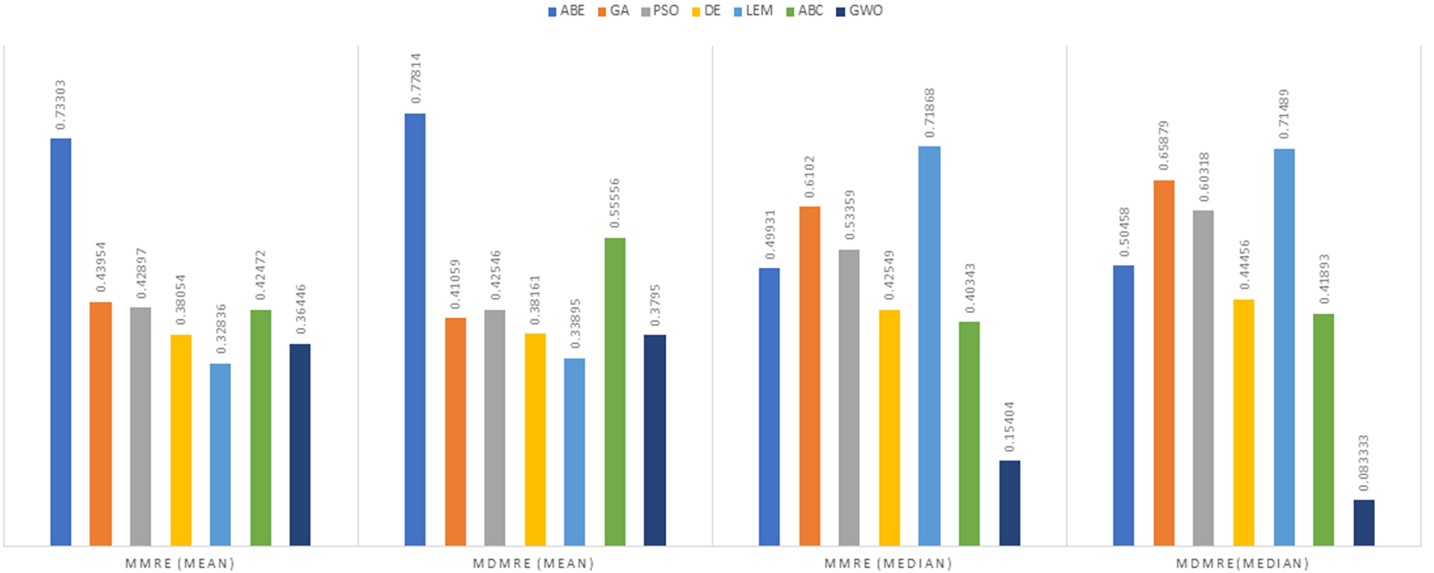

**Figure 7** Evaluation representation of MMRE and MdMRE criteria in the Albrecht datasets using the Euclidean similarity function.

## Desharnais dataset

According to the main results (Tables 6–9), when the Euclidean distance measure was applied, DABE exhibited the lowest MMRE (0.31061), closely followed by GWO (0.31265), suggesting superior prediction accuracy compared to other methods. For MdMRE, PSO-ABE demonstrated the lowest value (0.16667), indicating its proficiency in generating

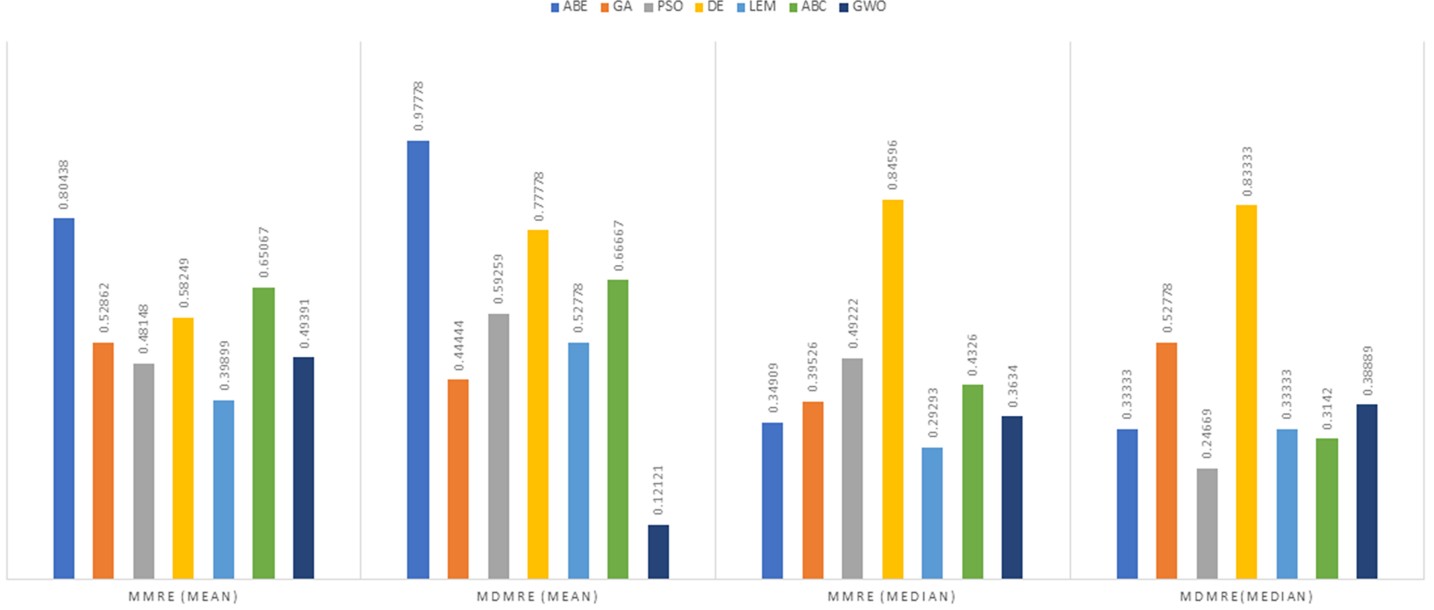

**Figure 8 Evaluation representation of MMRE and MdMRE criteria in the Desharnais datasets using the Manhattan similarity function.**

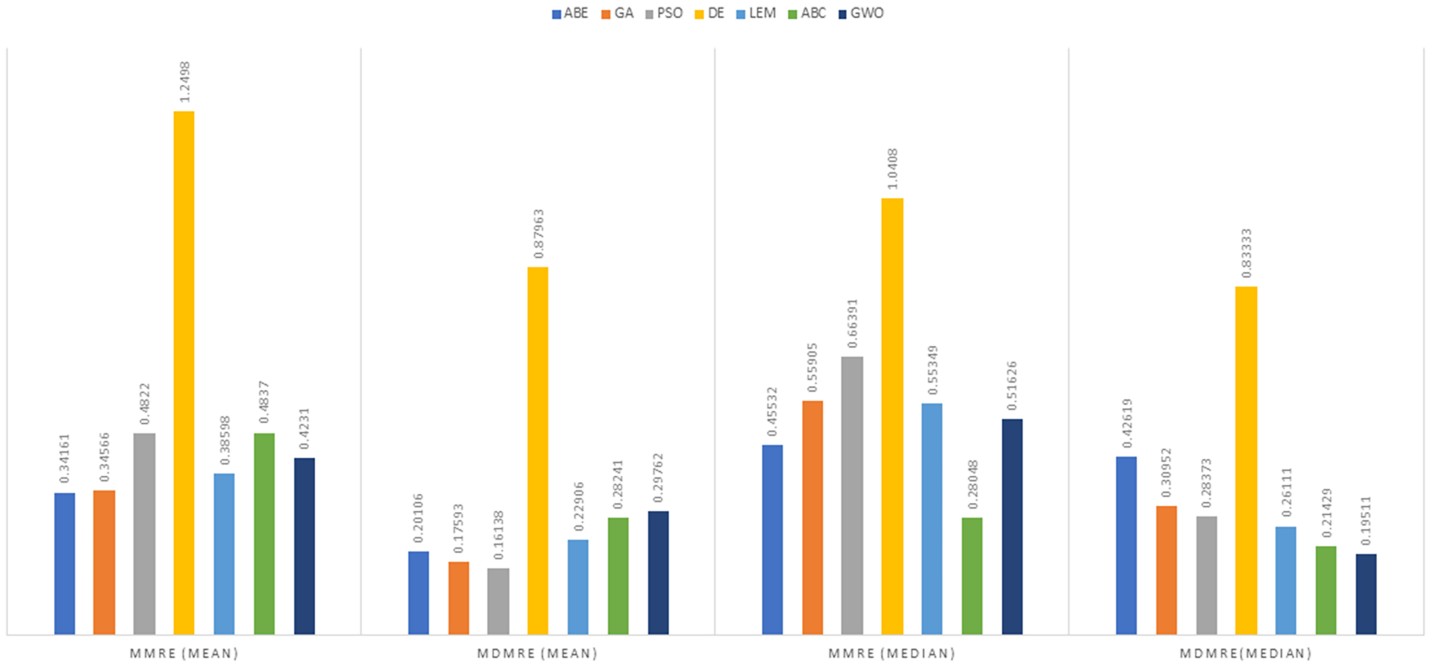

**Figure 9 Evaluation representation of MMRE and MdMRE criteria in the Maxwell datasets using the Manhattan similarity function.**

accurate predictions as well. Conversely, a higher PRED value signifies better prediction performance. ABC-ABE and GWO yielded the highest PRED values (0.5671 and 0.57319, respectively), showcasing their effectiveness in prediction. These observations imply that
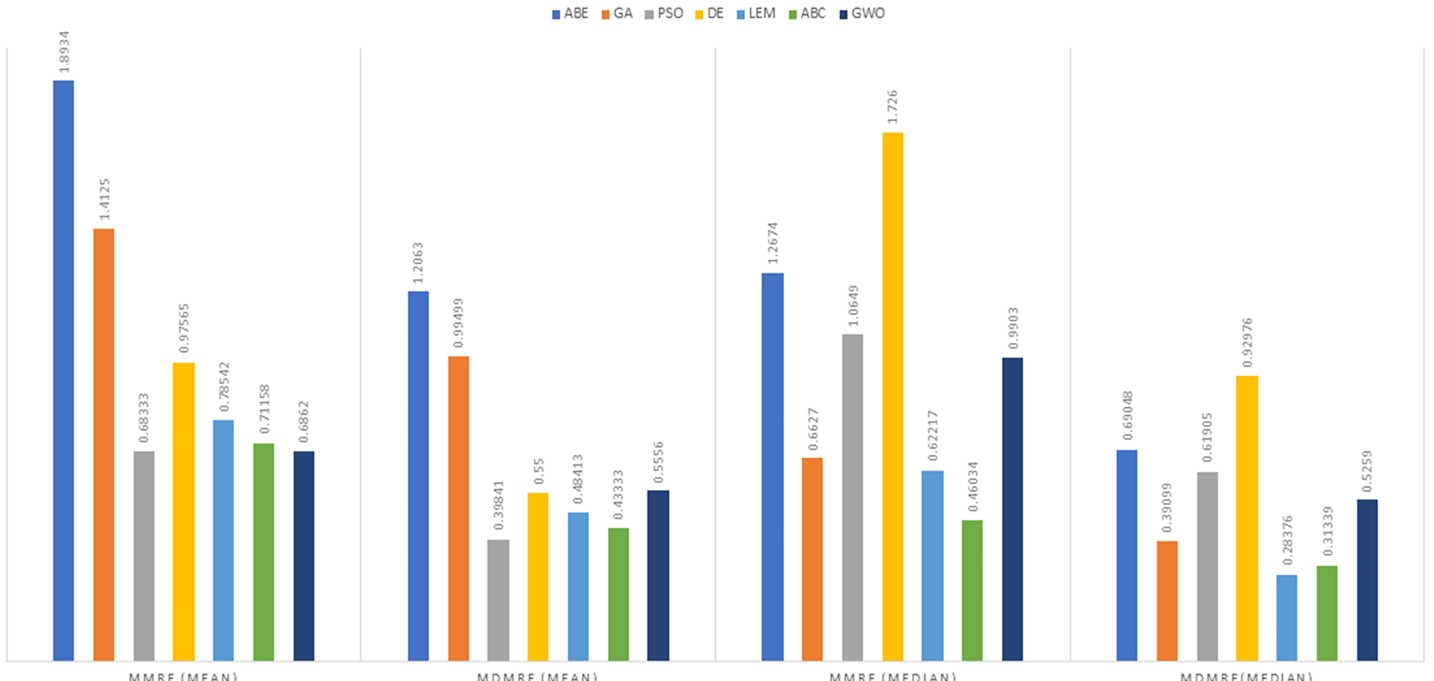

**Figure 10 Evaluation representation of MMRE and MdMRE criteria in the China datasets using the Manhattan similarity function.**

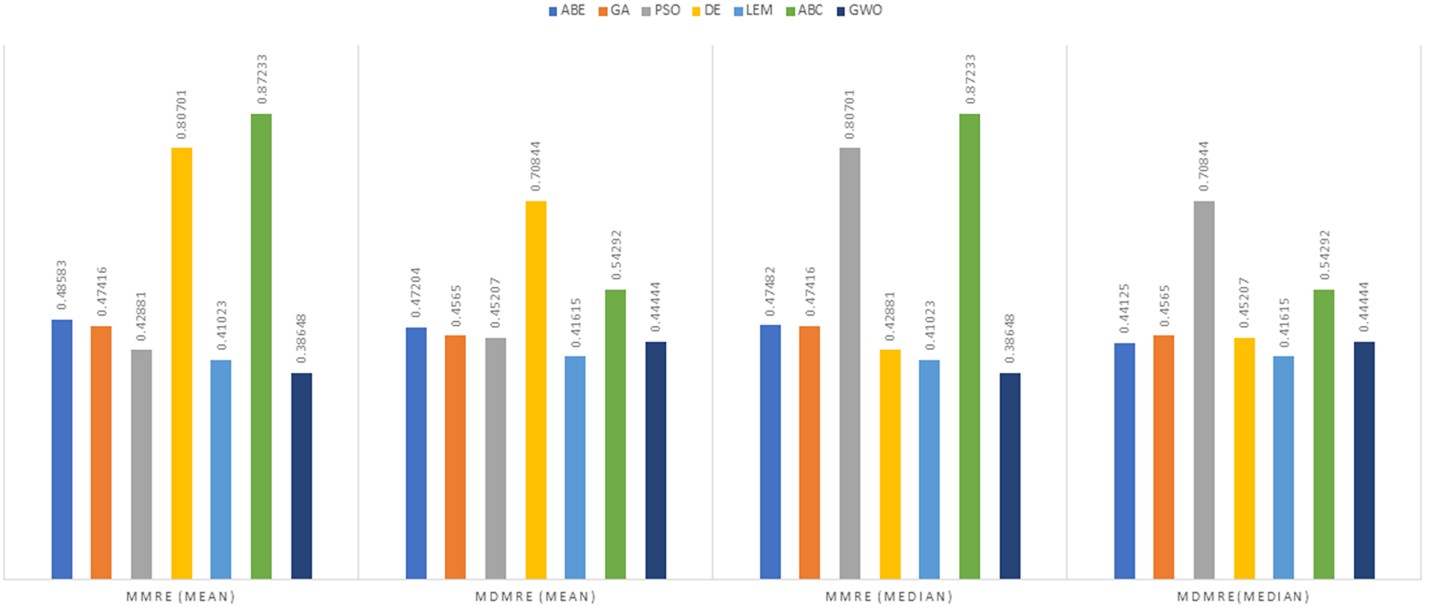

**Figure 11 Evaluation representation of MMRE and MdMRE criteria in the Albrecht datasets using the Manhattan similarity function.**

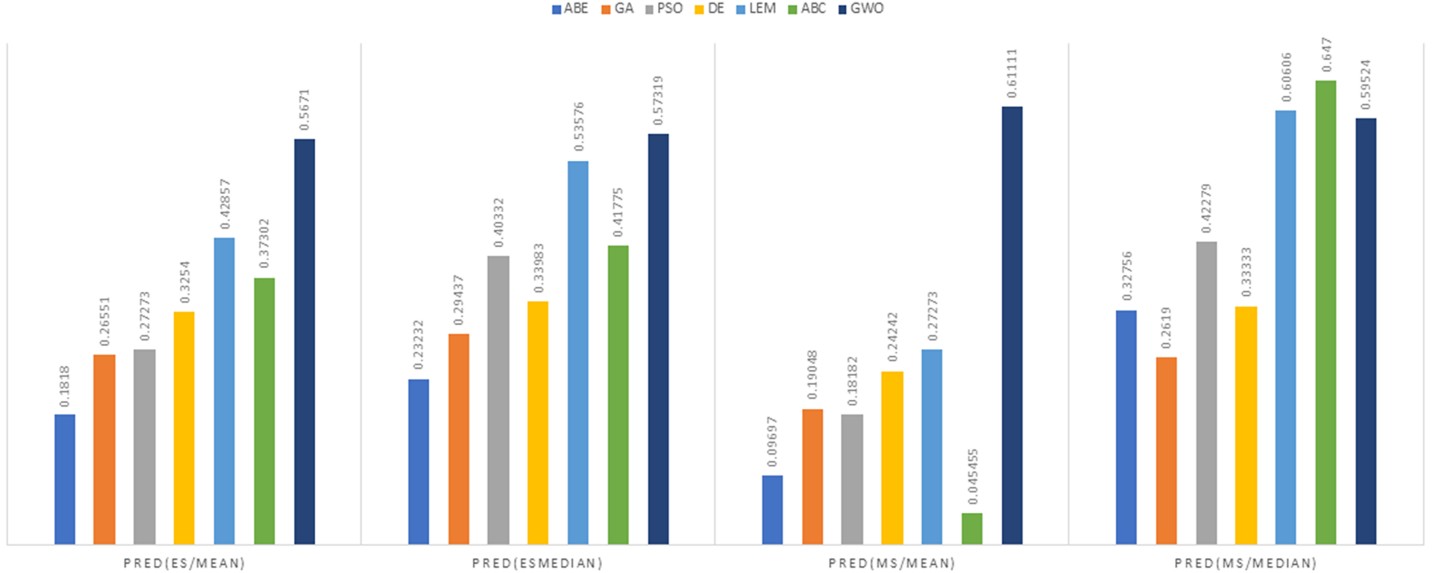

**Figure 12  PRED (0.25) criterion representation in the Desharnais dataset.**

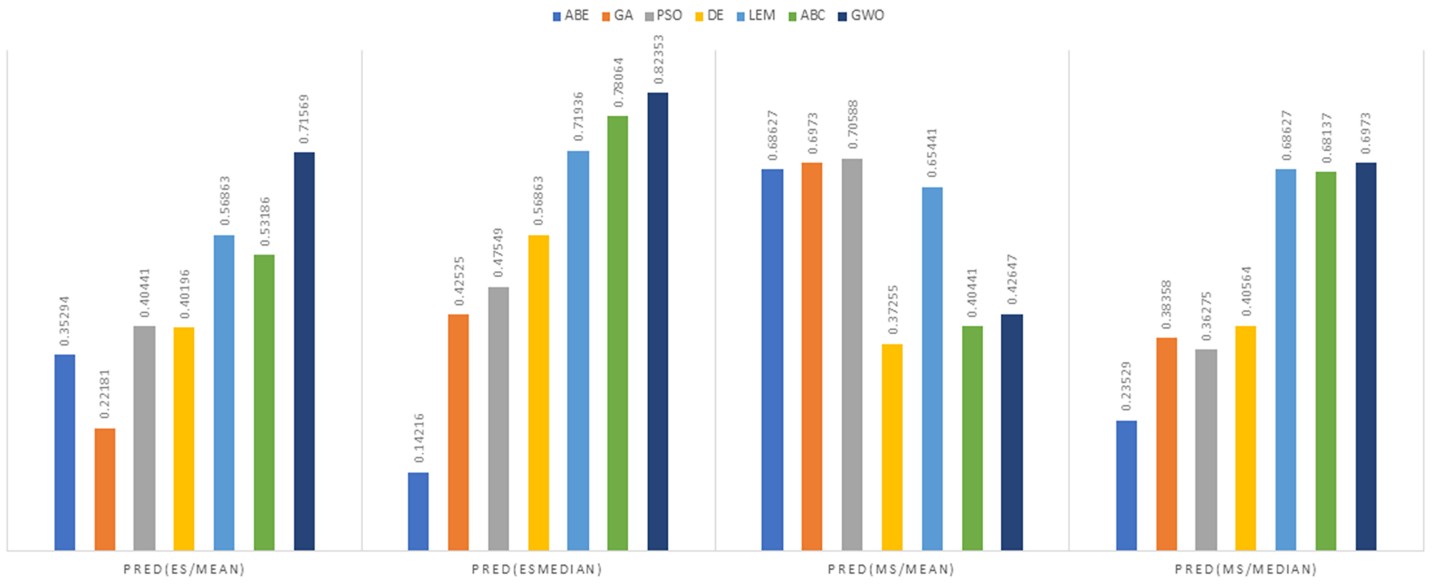

**Figure 13  PRED (0.25) criterion representation in the Maxwell dataset.**

DABE excels at reducing mean errors, PSO-ABE handles median errors proficiently, and ABC-ABE and GWO identify the largest share of acceptable predictions.

Under the Manhattan distance measure, DABE again showcased the lowest MMRE (0.39899), closely trailed by LEMABE (0.4326), suggesting their effectiveness in generating precise predictions based on the Manhattan distance measure. For MdMRE, GA-ABE and ABC-ABE demonstrated the lowest value (0.3142), followed by GWO (0.3634), indicating their proficiency in accurate predictions. ABC-ABE exhibited the highest PRED value
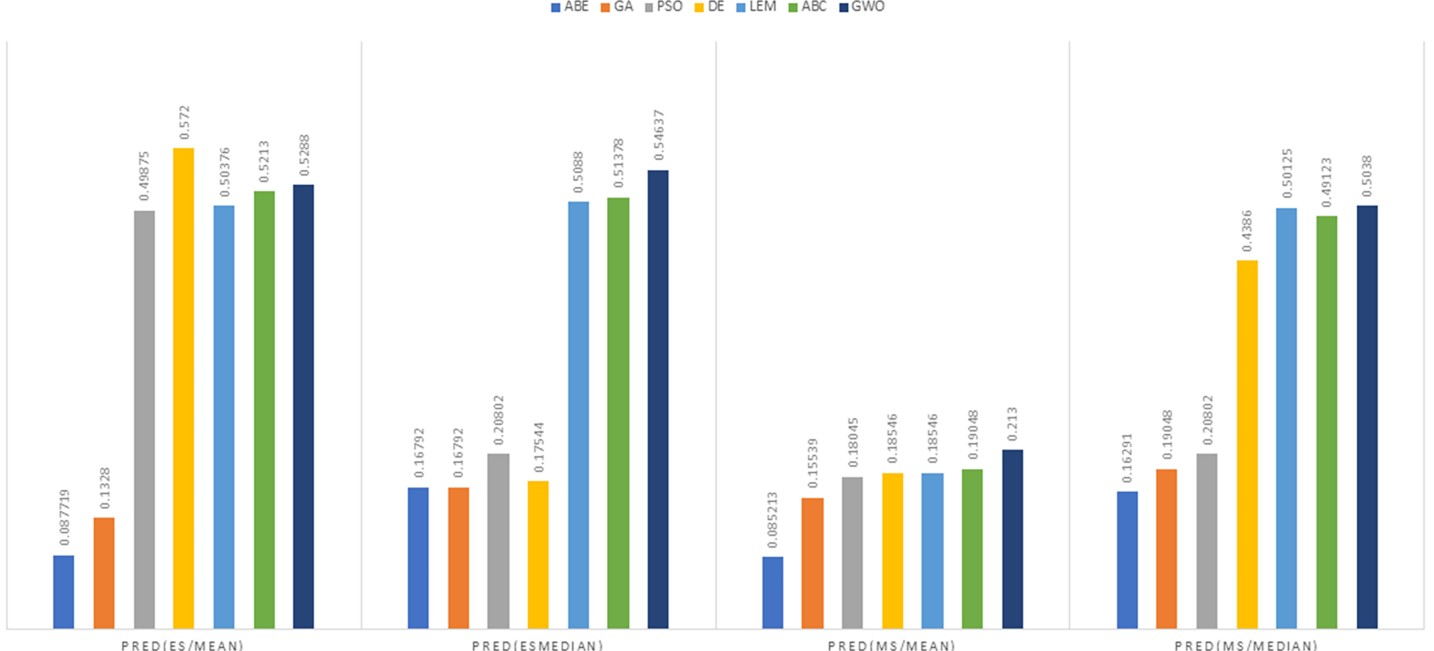

**Figure 14 PRED (0.25) criterion representation in the China dataset.**

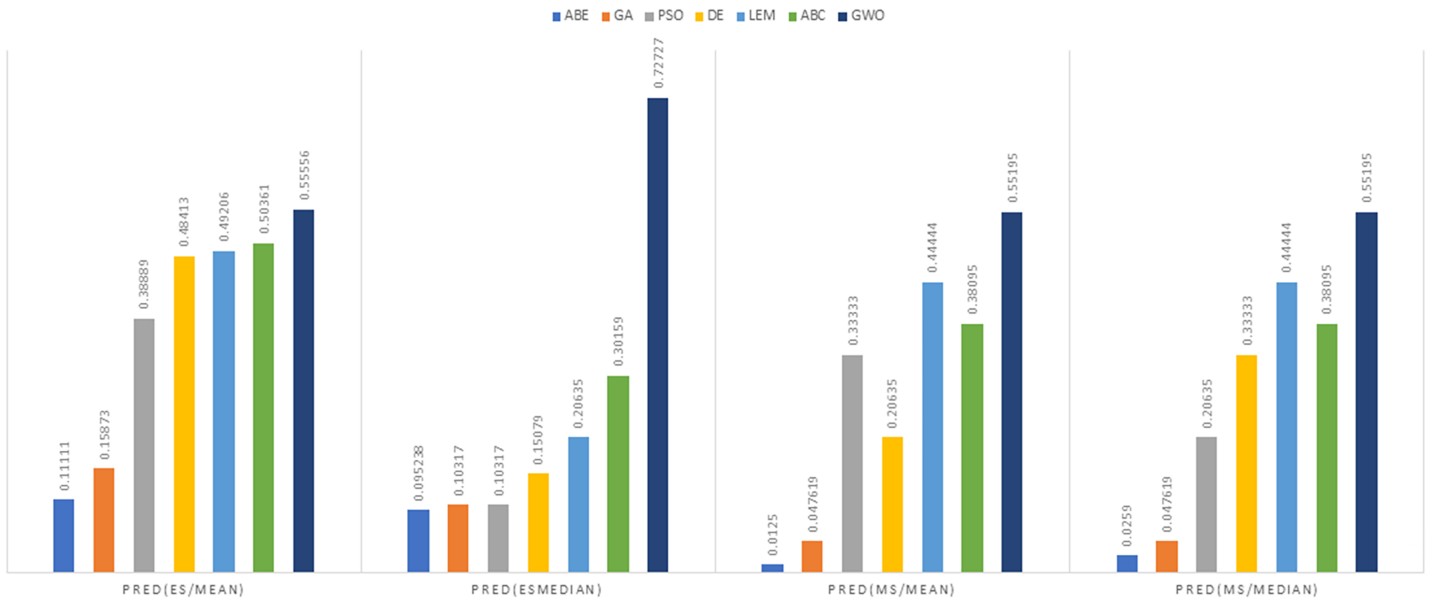

**Figure 15 PRED (0.25) criterion representation in the Albrecht dataset.**

(0.61111), closely followed by GWO (0.59524), showcasing their effectiveness in prediction according to this metric.

Overall, DABE and LEMABE excelled in MMRE, while GA-ABE, ABC-ABE, and GWO performed well in MdMRE and PRED.

## Maxwell dataset

In the context of the Euclidean distance measure, ABC-ABE stood out with the lowest MMRE value of 0.29997. Following closely is LEMABE with an MMRE of 0.45036, indicating its effectiveness in generating accurate predictions using this measure. About MdMRE ABC-ABE showcased the lowest MdMRE at 0.2254, closely trailed by LEMABE at 0.25833. These results underscore the proficiency of both algorithms in generating precise predictions based on the median. For PRED, GWO demonstrated the highest PRED value at 0.71569, closely followed by ABC-ABE at 0.53186. These results highlight the strength of GWO and ABC-ABE in terms of prediction according to this metric. A higher PRED value signifies better prediction performance, making GWO and ABC-ABE stand out in this regard under the Euclidean distance measure.

In the realm of the Manhattan distance measure, ABC-ABE exhibited remarkable performance, displaying the lowest MMRE at 0.28048, closely followed by LEMABE at 0.38598, showcasing their effectiveness in accurate predictions using this distance measure. ABC-ABE demonstrated exceptional proficiency, securing the lowest MdMRE of 0.19511, closely trailed by LEMABE at 0.21429, indicating their aptitude in generating precise predictions based on the median. ABC-ABE also emerged as a frontrunner, displaying the highest PRED value at 0.6973, highlighting its strength in prediction according to this measure under the Manhattan distance measure. While GWO still performed well in certain criteria, Maxwell highlights scenarios where ABC-ABE may edge out other approaches.

## China dataset

In the China dataset, when analyzing the results using the Euclidean distance measure, In the context of MMRE, GWO exhibited the lowest MMRE value of 0.6083, showcasing its remarkable predictive performance. Conversely, PSO-ABE displayed the poorest performance in MMRE with the highest value of 0.56695 among all methods. Shifting the focus to MdMRE, GWO again showcased the most impressive performance with the lowest MdMRE value of 0.3182. On the contrary, PSO-ABE displayed the least effective performance in MdMRE with the highest value of 0.33986, emphasizing its comparatively poorer predictive capability. Considering PRED, GWO outperformed all other methods with the highest PRED value of 0.5288, suggesting its strong predictive abilities. Conversely, GA-ABE exhibited the weakest performance in PRED with the lowest value of 0.1328, indicating its limitations in prediction according to this metric.

In the examination of results using the Manhattan distance measure, Within the domain of MMRE, GWO emerged as the frontrunner with the lowest MMRE value of 0.6862, denoting its exceptional predictive accuracy. Conversely, ABE demonstrated the least effective performance in MMRE, displaying the highest value of 1.8934, marking it as the weakest performer in this regard among all methods considered. Shifting the focus to MdMRE, GWO once again exhibited the most outstanding performance, boasting the lowest MdMRE value of 0.5556. Conversely, ABE showed the poorest performance in MdMRE, having the highest value of 1.2063, underlining its limited predictive capability based on the median. Considering PRED, GWO stood out with the highest PRED value of

**Table 10 Overview of best-performing methods per dataset and distance measure.**

| Dataset | Distance | Best method (MMRE) | Best method (MdMRE) | Best method (PRED (0.25)) |
|---|---|---|---|---|
| Desharnais | Euclidean | DABE (0.31061) | PSO-ABE (0.16667) | GWO (0.57319) |
| | Manhattan | DABE (0.39899) | GA-ABE & ABC-ABE (0.3142) | ABC-ABE (0.61111) |
| Maxwell | Euclidean | ABC-ABE (0.29997) | ABC-ABE (0.2254) | GWO (0.71569) |
| | Manhattan | ABC-ABE (0.28048) | ABC-ABE (0.19511) | ABC-ABE (0.6973) |
| China | Euclidean | GWO (0.6083) | GWO (0.3182) | GWO (0.5288) |
| | Manhattan | GWO (0.6862) | GWO (0.5556) | GWO (0.2130) |
| Albrecht | Euclidean | GWO (0.36446) | GWO (0.3795) | GWO (0.55556) |
| | Manhattan | GWO (0.38648) | GWO (0.44444) | GWO (0.55195) |

0.2130, affirming its robust predictive abilities. Conversely, PSO-ABE demonstrated the weakest performance in PRED, showcasing the lowest value of 0.085213, suggesting its limitations in prediction according to this metric.

### Albrecht dataset

In the evaluation of project cost estimation using the Euclidean distance measure for the Albrecht datasets, GWO exhibited superior predictive accuracy, achieving the lowest MMRE (0.36446) and MdMRE (0.3795) values, showcasing its proficiency in generating precise predictions, especially based on the median. Conversely, ABE displayed the highest MMRE (0.73303), indicating the least accurate predictions. Moreover, GWO demonstrated the highest PRED value (0.55556), implying robust prediction capabilities, while PSO-ABE exhibited the lowest PRED value (0.38889), suggesting relatively weaker prediction performance.

In the assessment of project cost estimation employing the Manhattan distance measure for the Albrecht datasets, GWO demonstrated superior predictive accuracy, displaying the lowest MMRE (0.38648) and MdMRE (0.44444) values, indicating precise median-based predictions. Conversely, PSO-ABE exhibited the highest MMRE (0.80701), signifying less accurate predictions. GWO also stood out with the highest PRED value (0.55195), denoting strong prediction capabilities, while ABE showcased the weakest predictive performance with the lowest PRED value (0.0125).

In light of these observations, Table 10 provides a concise overview of which method ranked best for each metric (MMRE, MdMRE, PRED) under Euclidean and Manhattan distance measures. While some algorithms such as ABC-ABE and DABE dominate specific metrics in particular datasets (*e.g.*, Desharnais or Maxwell), GWO stands out in the China and Albrecht datasets for most or all metrics.

As part of a deeper comparison, we computed how GWO-ABE's performance improves (or deteriorates) against each competing method (ABE, GA-ABE, PSO-ABE, DABE, LEMABE, ABC-ABE) under the Euclidean+Mean setting. We then averaged these percentage improvements across all four datasets. Tables 11 through 14 depict the results for each dataset in metrics MMRE, MdMRE, and PRED; Table 15 compiles the final overall improvement when summing up all comparisons.

**Table 11 Percentage improvement of GWO-ABE on Desharnais dataset.**

| Method | Imp. in MMRE (%) | Imp. in MdMRE (%) | Imp. in PRED (%) |
|---|---|---|---|
| ABE | +1.99% | −125.10% | +212.00% |
| GA-ABE | +14.36% | −50.00% | +113.60% |
| PSO-ABE | +33.06% | +5.26% | +107.90% |
| DABE | −0.66% | −38.47% | +74.20% |
| LEMABE | +8.26% | −28.57% | +32.30% |
| ABC-ABE | +35.93% | −20.00% | +52.00% |

As shown in Tables 11–14, each dataset reveals unique comparative insights when evaluating GWO-ABE against other ABE-based methods under the same configuration. In most cases particularly for Desharnais, China, and Albrecht GWO-ABE demonstrates significant reductions in MMRE and MdMRE, as well as increases in PRED(0.25). Nonetheless, there are instances (*e.g.*, with LEMABE in the Maxwell dataset or ABC-ABE in certain metrics) where the competing methods achieve lower errors or higher PRED values. These variations confirm that dataset characteristics, algorithmic design, and optimization strategies all contribute to the final performance.

To provide a consolidated view of how GWO-ABE compares with other ABE-based methods under identical conditions, we aggregated the percentage improvements in MMRE, MdMRE, and PRED(0.25) across all datasets and competing methods. These improvements were first computed for each dataset (Desharnais, Maxwell, China, Albrecht) relative to each algorithm (ABE, GA-ABE, PSO-ABE, DABE, LEMABE, and ABC-ABE), then averaged to yield a single value per metric. Table 15 compiles the final overall improvement when summing up all comparisons.

As shown in Table 15, GWO-ABE yields a moderate but consistent reduction in error rates, indicated by its positive average improvements of 15.63% for MMRE and 3.64% for MdMRE. More strikingly, its improvement in PRED(0.25) exceeds 100%, suggesting that, on average, GWO-ABE more than doubles the percentage of acceptable predictions compared to the other methods. This marked performance in PRED(0.25) can be attributed to the algorithm's capacity for assigning more suitable weights to key project features, thus improving the accuracy of identifying projects with similar attributes.

While certain methods occasionally surpass GWO in specific metrics or datasets, the summarized findings suggest that GWO-ABE remains highly competitive overall. It achieves moderate reductions in average error (MMRE and MdMRE) and exhibits a substantial increase in PRED. Consequently, GWO-ABE appears to be a particularly promising enhancement to analogy-based cost estimation, combining robust error minimization with a high likelihood of generating practically usable predictions.

## LIMITATIONS

The present research investigates how software effort estimation can be improved using an analogy-based approach and GWO algorithm; however, this method has some limitations that must be considered.

**Table 12  Percentage improvement of GWO-ABE on the Maxwell dataset.**

| Method | Imp. in MMRE (%) | Imp. in MdMRE (%) | Imp. in PRED (%) |
|--------|------------------|-------------------|------------------|
| ABE | +24.06% | +40.63% | +102.80% |
| GA-ABE | +29.23% | +40.86% | +222.60% |
| PSO-ABE | +43.80% | +61.70% | +77.00% |
| DABE | +5.73% | +9.39% | +78.00% |
| LEMABE | −19.20% | +12.75% | +25.86% |
| ABC-ABE | −78.96% | +21.12% | +34.57% |

**Table 13  Percentage improvement of GWO-ABE on the China dataset.**

| Method | Imp. in MMRE (%) | Imp. in MdMRE (%) | Imp. in PRED (%) |
|--------|------------------|-------------------|------------------|
| ABE | +58.90% | +63.61% | +503.00% |
| GA-ABE | +45.73% | +51.46% | +298.00% |
| PSO-ABE | −7.29% | +6.37% | +6.03% |
| DABE | +40.25% | −115.20% | −7.56% |
| LEMABE | +34.93% | +45.40% | +4.97% |
| ABC-ABE | +15.17% | +16.30% | +1.44% |

**Table 14  Percentage improvement of GWO-ABE on the Albrecht dataset.**

| Method | Imp. in MMRE (%) | Imp. in MdMRE (%) | Imp. in PRED (%) |
|--------|------------------|-------------------|------------------|
| ABE | +50.30% | +51.20% | +400.00% |
| GA-ABE | +17.08% | +7.58% | +250.00% |
| PSO-ABE | +15.05% | +10.80% | +42.86% |
| DABE | +4.23% | +0.55% | +14.76% |
| LEMABE | −11.00% | −12.00% | +12.90% |
| ABC-ABE | +14.18% | +31.70% | +10.31% |

**Table 15  Overall improvement (%) of GWO-ABE across all comparisons.**

| Metric | Average improvement (%) |
|--------|-------------------------|
| MMRE | +15.63 |
| MdMRE | +3.64 |
| PRED (0.25) | +111.23 |

One major limitation of this study, similar to other research in software development effort estimation, is the reliance on a small number of publicly available standard datasets. These datasets were chosen for their relevance and widespread use in prior studies, as they provide a reliable benchmark for comparing estimation models (*Fávero, Casanova & Pimentel, 2022*; *Tawosi, Moussa & Sarro, 2022*). However, their limited size and scope may not fully capture the diverse and complex nature of software development projects (*Li et al., 2024*). To mitigate this limitation, future research can explore alternative datasets that cover a broader range of projects, industries, and development methodologies.

Additionally, dataset replication techniques, such as synthetic data generation or augmentation strategies, could be employed to create more diverse and representative datasets while preserving the integrity of real-world data distributions. Collaborating with industry partners to access proprietary datasets under confidentiality agreements may also enhance the robustness and applicability of software effort estimation models.

Unlike recent studies leveraging large-scale industrial datasets, such as Deep-SE and GPT2SP, which analyzed datasets containing over 23,000 issues, this research is constrained by the lack of access to proprietary datasets due to privacy and organizational restrictions. Consequently, while the datasets used here are well-established and suitable for evaluating the proposed model, their smaller size may restrict the generalizability of the findings.

The specific characteristics and contexts of these datasets may lead to a model that performs well on these particular datasets but may not generalize effectively to other datasets with differing attributes or complexities. Further validation using a broader range of datasets, particularly large-scale industrial data, is necessary to enhance the robustness and applicability of the proposed method.

Another important limitation is that there are no comprehensive features relating to human factors in the existing datasets used. Factors such as team experience, individual productivity, and management practices significantly impact software effort estimation. Nevertheless, due to a lack of detailed information regarding these factors in these sets, the accuracy and reliability of the effort estimates are limited. In future studies, this limitation could be overcome by providing more information about human factors through questionnaire surveys and supplementary metadata or qualitative data derived from interviews or questionnaires.

## MANAGERIAL INSIGHTS

This study highlights several key insights for practitioners, decision-makers, and policymakers in the software industry and similar domains. These insights aim to bridge the divide between theoretical advancements and practical applications, facilitating informed decision-making and strategy development:

*Optimized cost estimation with GWO:* By integrating optimization algorithms like GWO into ABE methods, organizations can achieve more accurate and reliable cost estimates. This minimizes the risk of budget overruns and project setbacks, which are common challenges in software development (*Ali, Ren & Wu, 2024*). The GWO's ability to dynamically adjust feature weights and balance exploration and exploitation makes it particularly effective for industries relying on data-driven project estimations.

*Practical adoption of enhanced ABE methods*: Implementing enhanced ABE methods requires access to high-quality historical datasets and a focus on feature optimization. Organizations can adopt these methods to improve project planning, resource allocation, and risk management. Furthermore, training teams to understand and utilize these techniques can ensure smoother adoption and better integration with existing workflows (*Nevena Rankovic, Ivanovic & Lazić, 2024*).

*Leveraging data-driven approaches:* High-quality data is the foundation of effective cost estimation. Decision-makers should prioritize collecting, curating, and maintaining standardized datasets to enable reliable predictions. The insights from this study emphasize the importance of using structured and comprehensive data for improving accuracy in cost estimation models, which can also be applied across various sectors beyond software development.

*Policy recommendations for data standardization:* Policymakers have a critical role in promoting the adoption of advanced cost estimation techniques. Encouraging the standardization of data formats and collection processes across industries can significantly enhance the generalizability and scalability of such models. Additionally, fostering collaborations between academia and industry for data sharing can provide richer datasets, enabling further advancements in estimation accuracy.

*Strategic planning and budgeting:* For managers and decision-makers, this study highlights the strategic value of incorporating AI-driven cost estimation models into budgeting practices. By leveraging these models, organizations can identify potential risks early, allocate resources more effectively, and improve overall project outcomes. These benefits are particularly relevant for industries dealing with complex, large-scale projects where traditional estimation methods often fall short.

By providing these insights, this study not only enhances the understanding of advanced cost estimation techniques but also offers actionable guidance for stakeholders to implement these methods in practice, ensuring better decision-making and project success.

## CONCLUSION

This research introduces an innovative method for SCE by combining the GWO algorithm with the ABE technique. The findings highlight the efficacy of the proposed GWO-based ABE approach, which outperformed ABE-customized methods across multiple datasets and evaluation metrics. Compared to such ABE methods like ABC-ABE and PSO-ABE, the GWO-based ABE achieved up to 15% reduction in mean MMRE and 111% improvement in PRED (0.25). These enhanced conclusions can be attributed to the GWO algorithm's ability to optimize the feature weights in the similarity function, a crucial component of the ABE method.

The integration of the GWO algorithm addresses the limitations of customized-ABE methods and leverages the strengths of metaheuristic optimization to provide more robust and efficient SCE. These results advocate for the GWO-based ABE as a promising approach to enhance the accuracy and reliability of software project planning and management. However, it is essential to acknowledge that these conclusions are based on the analysis of the four datasets employed in this study. The experimental results indicate that GWO-ABE achieved the highest accuracy on the China dataset, with the most significant improvement in PRED, while the lowest accuracy was observed on the Maxwell dataset, where the model exhibited a decrease in MMRE and limited improvement in PRED. While these datasets were chosen for their relevance and consistency with related

research, further validation on a broader range of datasets is needed to confirm the generalizability of the findings.

Future research directions may explore the application of other optimization algorithms and investigate the impact of dataset characteristics on the performance of the proposed method. For instance, algorithms such as the whale optimization algorithm (WOA) or cuckoo search algorithm could be examined for their potential to enhance feature weighting and similarity measurement in ABE methods. Overall, this study adds to the expanding body of work focused on advancing SCE by leveraging the combined strengths of machine learning and optimization methods.

### Funding
This work was supported by Universiti Putra Malaysia. There was no additional external funding received for this study. The funders had no role in study design, data collection and analysis, decision to publish, or preparation of the manuscript.

### Grant Disclosures
The following grant information was disclosed by the authors:
Universiti Putra Malaysia.

### Competing Interests
The authors declare that they have no competing interests.

### Author Contributions
- Taghi Javdani Gandomani conceived and designed the experiments, performed the experiments, authored or reviewed drafts of the article, and approved the final draft.
- Maedeh Dashti conceived and designed the experiments, performed the experiments, analyzed the data, prepared figures and/or tables, and approved the final draft.
- Sadegh Ansaripour analyzed the data, performed the computation work, prepared figures and/or tables, and approved the final draft.
- Hazura Zulzalil performed the experiments, authored or reviewed drafts of the article, and approved the final draft.

### Data Availability
The datasets and the simulation source code are available in the Supplemental Files.

### Supplemental Information
Supplemental information for this article can be found online at http://dx.doi.org/10.7717/peerj-cs.2794#supplemental-information.

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
