# Peer review of "Enhancing analogy-based software cost estimation using Grey Wolf Optimization algorithm"

_PeerJ Computer Science, doi:10.7717/peerj-cs.2794_

## Round 0.1 · original submission · Major Revisions

Dear authors,

Thank you for the submission. The reviewers’ comments are now available. It is not suggested that your article be published in its current format. We do, however, advise you to revise the paper in light of the reviewers’ comments and concerns before resubmitting it. The followings should also be addressed:

1. Information about the datasets should be provided in the Abstract section.
2. Pay special attention to the usage of abbreviations. Spell out the full term at its first mention, indicate its abbreviation in parenthesis and use the abbreviation from then on.
3. The motivation and reason of using Gray Wolf Optimization among many other metaheuristic algorithms for the focused problem should be mentioned.
4. Equations should be used with correct equation number. Many of the equations are part of the related sentences. Attention is needed for correct sentence formation. Equation numbering should be corrected.
5. All of the values for the parameters of all algorithms should be given.
6. Encoding type or representation scheme and fitness function of Gray Wolf Optimization should be provided. How constraints for Gray Wolf Optimization (for example in decision variables intervals) are handled is not clear.
7. Some of the paragraphs (for example in the Discussion section) are too long to read. They should be divided into two or more for readability and comprehensibility.

Best wishes,

Reviewer 1 ·

Basic reporting

In this paper, authors present an application of the Gray Wolf Optimizer to software cost estimation, more specifically, to enhance the analogy-based estimation broadly use in the software industry. To this end, authors introduce their investigation with some statistics from the mentioned industry and take into consideration novel machine learning methods that have been previously used for software cost estimation.

1. I would like to recommend the authors to carefully proofread their manuscript, because there are many typos and grammar errors throughout the text. I am sorry I cannot list all the mistakes I found.

2. Authors present vaguely what ABE in software cost estimation is. We have to remember that there are more than one method for cost estimation and not all readers would be familiarized with all the terms used. For example, what is:
- function point
- lines of code
- degree of similarity
- nearest neighbor

3. In this respect, the notation used in equations (1) and (2) is incomplete and ambiguous and they must be corrected. For example:
- What is p.p in sim(p.p)?
- Is F equals to f?
- The Dis function is missing in the cases
- Is W equals to w?

4. In these equations, function Dis only considers features that are numeric, but what if they are not numeric? How can Dis be calculated? Continuing in the similarity functions, how can we compare software features?

5. Regarding the KNN, what does k refers to?

6. Regarding the related work, we must remember that the objective of the literature review is know what has been done and what are the gaps which lead to opportunities for contribution to knowledge. I must say that this literature review fails in this respect, because authors do not analyze what they have found. It is not enough to list and comment a large amount of previous papers, but we also need to know that the work presented here is, actually, a real contribution. For example, authors state in line 330:

"As it seems using AI and machine learning algorithms has improved ABE, we use the gray wolf algorithm in the feature weighting method in this study, considering its main features and advantages."

But I think that the use of GWO was not justified after the literature review. What I mean is, why GWO and not PSO, ACO, GA, or other bio-inspired method?

Moreover, let us remember the No-Free Lunch theorem, thus, what is the need for yet another algorithm for supporting ABE?

7. Line 341 should be a subsection.

8. In line 358, what should be inferred with "similar projects"? Does this refer to equations (1) and (2)? How is the projects similarity based on non-numeric features could be quantified?

Experimental design

In their experimental study, authors used three well-known performance metrics, namely MMRE, MMRE, and PRED, and used four public datasets for experimentation.

9. Are four datasets enough for making conclusions? I think four datasets are not enough and that is why authors mention this subject as a limitation. Then, authors should not make general conclusions, but only consider these four datasets.

10. More importantly, only the basic characteristics of the four datasets are shown, but the projects behind these datasets are not presented. What are the features of these projects?

11. In line 445, authors state that

"To design the experiment, we initially review different combinations and finally select the fittest resulting model to be compared to other algorithms."

What is the meaning of this? It seems like it is a parameter tuning step, so, is it fair to tune parameters only for the proposed approach?

12. Were all other algorithms used for comparison implemented by the authors?

Validity of the findings

13. Considering the results of all the tables, we can see that the proposed GWO-ABE does not find the best solutions in all experiments. Moreover, it only finds the best solutions in a few experiments. Then, what is the real contribution of this study?

14. It is not clear what the figures in these tables are results of. We must remember that GWO is not a deterministic algorithm and should be run many times in order to obtain statistical results. How many times was it run for each instance?

15. Furthermore, after it is run many times, statistical test have to be applied. Did authors run any statistical test on their results for statistical significance?

Additional comments

After all my comments, conclusiones are not completely supported.

·

Basic reporting

-Expanding the Scope of Research: It may be beneficial to test the proposed method on a larger dataset or different types of projects.
-Deeper Analysis: The paper could include a deeper analysis of the results of using different feature weights and their impact on accuracy.

Experimental design

no comment

Validity of the findings

no comment'

Additional comments

no comment'

·

Basic reporting

The manuscript is well structured and presented. However, there are rooms for improvement.

- In case of specific name, capital letter should be used. For example, "particle swarm optimization" in line no. 86. Likewise, "gray wolf optimization" in line no. 188 and "honey bee colony" in line no. 324.

- In the introduction section, why was the GWO chosen for this work. I suggest that you
add theoretical and/or practical reasons at lines 85-88.

- In line no. 187, Gray Wolf Algorithm (GWA) or Gray Wolf Optimization (GWO).

- In the related work section, the presentation of the literature review on the applications of AI algorithms should be improved. Previous research works have used metaheuristics to improve the optimization performance. I suggest that you should provide an additional table to summarize the algorithms used, problem characteristics, optimization functions precision estimation metrics, evaluation criteria, datasets used, and so on. The additional table may be similar to Table 1.. This could help to substantiate another contribution in this work by fulfilling a research gap in the field.

- Citation style should be improved. It is repetitive. For example in line no. 284, "In 2006, Huang and Chiu (Huang & Chiu, 2006) concluded that ....". Another example in line no. 289, "in 2009, Li et al. (Y.-F. Li, Xie, & Goh, 2009).

- Updating literature review since the review was limited up to 2022.

Experimental design

- In Table 4-7, I suggest that the best solutions obtained from each dataset should be highlighted using either bold number or underline number, etc. This will help reader to easily spot the best results obtained from each method.

- What were the execution times taken by the AI algorithms? How long did it take to find the solution?

- In Figure 4-15, grey-scale colors do not help for presenting the graphs. I suggest that normal colors will be better to depict the graphs.

Validity of the findings

- The performance of the AI algorithms usually depend on its parameter setting. What were the parameter settings for those AI methods? Why? Where do they come from?

- Regarding the comparison between AI algorithms, the experimental design to promote fair comparison needs to be clarified. What was the design for fair comparison?

- I suggest that statistical tests should be applied to identify the statistical significant on the results obtained from the proposed algorithms.

Additional comments

- Sensitivity analysis should be additionally presented as a subsection. The analysis will help reader to understand how responsive the output is to change in certain variables or parameters.

- I suggest adding a subsection of managerial insights before conclusion section by listing the insight issues for users, decision makers and/or policy makers for similar industries.

·

Basic reporting

(49-53) Can it be replaced with the latest survey results? Most of the research results on SEE refer to the survey, I recommend adding the latest survey results, which will later be a reference for future research
(56-58) Could additional references and paragraphs be added that corroborate this opinion? Of course, the latest and relevant references. Although in the paragraph after that, you reveal the paragraph of ABE's advantages, I recommend that you first give an opinion of the previous research whose results show that ABE is superior.
(174) You can add an opening paragraph to explain this Solution Function, including a clear definition and general use. I also recommend adding references to give the impression that you are using a strong theory, especially for estimating software effort.

Experimental design

(104-105) Can appropriate additional references and paragraphs be added? Of course, the latest and relevant references. Thus, it will add to the strength and confidence of the ABE method. I also don't know. Is it okay to immediately discuss "ABE in software cost estimation" after the "introduction" without adding a special chapter, "Method" or "Methodology"? You can check the journal template again.
(123-125) Can additional references and paragraphs be added? Of course, the latest and relevant references. Here, you choose a specific to measure the distance. Why not opt for other distance methods such as Chebishev, Canberra, or Chi-Square? With this explanation, it will be easier for future research to replicate or make improvements.
(344-345) The division into three parts of the dataset is better to clearly state the number of percentages for each dataset. I suggest you explain another reason why you divide the data by the percentage you choose?
(409-410) Describe any previous studies that used each of the datasets you used. This also makes it easier for further research to replicate.
(411-412) Please be consistent in writing quotes. I recommend paraphrasing each paragraph you quote. Author inconsistency affects article quality.
(422-423) Reasons for not using the ISBSG Dataset: please give clear and detailed reasons. If possible, give reasons from other studies for not using ISBSG, especially since more SEE studies also use ISBSG. If ISBSG is not used in this study, I recommend deleting this part.
(427-428) Provide references to research that uses preprocessing like this. You can add some illustrations or formulas about this preprocessing. I recommend that you provide additional paragraphs of paraphrasing from other research results cited.
(591) Are you serious about using regression methods? Is this the result of GenAI that has not been paraphrased? I'm sure not. I recommend that you review your writing over and over again before presenting it to others.

Validity of the findings

(453-457) Can you explain whether the research baseline you are using is as described in the related work section? If not, I recommend readjusting the related work section so that the flow of this research is appropriate and continuous.
(598-560) Which dataset can you add from that doesn't describe the feature you are referring to? I suggest that clarity in defining the limitations of research will be filled by the next research by referring to the results of your research.
(610-612) If, in your conclusion, you present the percentage decrease and increase, I suggest that you add a new table containing the results of the percentage comparison of each method. This makes it easier to draw appropriate conclusions between the research problem and the research results.
(618-619) You can add an example of the algorithm you are referring to for future research. This adds insight for future research.

Additional comments

Good research, recommended to continue if it meets the review results.

---

## Round 0.2 · Major Revisions

Dear Authors,

It is the opinion of two of the reviewers that the concerns have not been addressed with sufficient clarity, and that the necessary additions and modifications have not been performed. The paper therefore still needs to be revised. It is strongly recommended that the reviewer comments are given due attention, and that the necessary arrangements for the article are fulfilled with the utmost care.

Best wishes,

Reviewer 1 ·

Basic reporting

Authors have attended adequately all my previous concerns.

Experimental design

Authors have attended adequately all my previous concerns.

Validity of the findings

Authors have attended adequately all my previous concerns.

Additional comments

Authors have attended adequately all my previous concerns.

·

Basic reporting

1. Regarding my previous comment on the case of specific name, capital letter should be used. The authors have responded to this comment that “We have carefully reviewed the manuscript and ensured that all abbreviations are correctly introduced”. However, this is not true. For example, "particle swarm optimization" in line no. 91. Likewise, "artificial bee colony " in line no. 400. I am sorry that I could not list all of them.
2. Again, my previous comment on the citation style should be improved. The authors have responded that “we have corrected it”. However, this is not true. The citation style still needs to be improved. There are many of those and I am sorry that I could not list all of them. For examples:
- Line no. 81, “The combination of models has to comply with a set of fusion rules recommended by (Wen et al. 2012)”. It should be “The combination of models has to comply with a set of fusion rules recommended by Wen et al. (2012)”.
- Likewise in line no. 399, “(Dashti et al. 2022) investigated the learnable evolution model (LEM), and (Shah et al. 2020) investigated …”. It should be “Dashti et al. (2022) investigated the learnable evolution model (LEM), and Shah et al. (2020) investigated …”.
- Line no. 628, “The convergence parameters (a, A, and C) followed the standard values described in [13]”. What is [13]? In the References section, “[13]” is not listed.

Experimental design

3. The comparison on the execution times taken by those AI algorithms must be academically reported in this work. Due to the problem domains, parameter settings, limitations and constraints, experimental design, objective functions and program coding; the performance efficiency of the algorithms could be the same or different from the previous research works. That is why the comparison on the execution times must be reported in this work. The discussions on the algorithms’ performance compared with the previous works can be provided later on.

4. The performance of the AI algorithms usually depend on its parameter setting. What were the parameter settings for those AI methods? Why? Where do they come from? The authors have responded to my previous comment on line 626-627 that “For GWO, the population size was set to 8 and the number of iterations to 200, based on recommendations from prior research.” Which one? The statement on line 627-628 stated that “The convergence parameters (a, A, and C) followed the standard values described in [13], …”. However, the reference number [13] has not been mentioned in the reference section, which is not formatted in numerical style. Again, what were the parameter settings for all methods?

5. Regarding the performance comparison between AI algorithms, the experimental design to promote fair comparison needs to be clarified. What was the design for fair comparison on algorithms’ performance? The authors have responded to this comment based on the problem setup using the same datasets, each of which was based on 30 runs. What was the explanation on the algorithms setup to promote the performance comparison between AI algorithms?

Validity of the findings

6. The statistical tests help to identify the statistical significance of the computational results obtained from the proposed method compared with other algorithms. It confirms that the proposed method outperformed the other algorithms based on the statistical significance.
Since the GWO algorithm was introduced in 2014, there are more than 10,000+ published articles in SCOPUS database. The demonstrations on the robustness and reliability of GWO have been widely published to solve various problem domains. A few of those published articles were related to software cost estimation.

·

Basic reporting

(57-58) You can mention in advance some of the methods used in the SEE, and many systematic literature studies should also discuss them. You can use the study results to start discussing the specifics of this research.
(58-59) Before going into the paragraph about ABE (as the basic method used), you can provide other algorithmic or non-algorithmic methods in SEE, such as Use Case Point, Function Point, Planning Pocker, etc.
(87) You can tell us about Computational Intelligence or Bio-inspired optimization before entering GWO from general to specific explanations.
(151-153) Provide a reference stating that Euclidean and Manhattan are distance methods widely used in some previous studies for the similarity function.
(219) It would be better to clarify the example: FPs = Function Points or others.
(246) The writing of the citation or paragraph "Mirjalili et al. (Mirjalili et al. 2014) in 2014" may be corrected if it is incorrect. So that readers can understand more easily.
(46, 115, 263, 340, 460, 511, 701) The words "Portions of this text were previously published as part of a preprint (Gandomani et al. 2024a)." are the same used at the beginning of each paragraph. Is this a mistake, or how? It can be used as a reference for the next revision. If this is automatically from the system, you can ignore it.
(264) You can add about Computational Intelligence or Bio-inspired optimization before entering GWO. From general explanations to specific explanations
(549) To "write citations" in this paper in APA or IEEE format? Is this just a typo or a misquotation? Author inconsistency affects article quality
(628) To "write citations" in this paper in APA or IEEE format? Is this just a typo or a misquotation? Author inconsistency affects article quality

Experimental design

(235) You choose k=3. Maybe you can mention the reason and reference
(515) Why not use MAE, MSE or R2 as is used in many SEE studies? This is an explainable reason
(581) Create a new help table that displays the settings for all parameters (variables and their values) used in the design of this study. The table is more straightforward to add to the understanding and be replicated.
(596) Provide references that also use 3-fold cross-validation in SEE

Validity of the findings

(789) To give a little input on the limitations encountered in this study, for example, if the existence of public data is the reason, what can be done? Replace the dataset or perhaps create a dataset replication.
(862) The conclusion that GWO-ABE can reduce MMRE by 25% and increase PRED by 18% has not been explained where it came from. The findings are at the heart of this study. Please provide a table or other illustration explaining these findings.

Additional comments

Hopefully it can be revised according to the needs and future of research in the field of SEE.

---

## Round 0.3 · Minor Revisions

Dear Authors,

Thank you for addressing the reviewers' comments. Although one reviewer accepts your paper, one reviewer suggests minor revision. We strongly recommend that you address the issues raised by Reviewer 4 and resubmit your paper after making the minor changes.

Best wishes,

·

Basic reporting

-

Experimental design

-

Validity of the findings

-

Additional comments

The quality of the revised manuscript has improved a lot. Therefore, it can be accepted for publication.

·

Basic reporting

(61) Referring to Rashid (2023), FPA and UCP are Algorithmic (https://ieeexplore.ieee.org/stamp/stamp.jsp?arnumber=10243029)
(63-65) This sentence may not be right for its position, because the next sentence you directly explain ABE, not discussing ML or Optimization. Maybe you can move it before line 79.
(91-92) You can add a comparative reference to several other ML Methods such as the results of this study (https://ieeexplore.ieee.org/document/10468186)
(100) The abbreviations of GA and PSO are written first, then the abbreviations, for example Genetic Algorithm (GA).
(274) Before entering the points of project similarities, an opening sentence can be added to explain the points that will be discussed below.

Experimental design

(492) Data splitting 60, 20, 20, it seems like 20% is used in Model Training. Is that what it means? If yes, this paragraph can be distinguished from Model Training and Model Testing, use a new paragraph for Data Splitting.
(564) Can you provide references to several papers that use PRED 0.25 and 0.30 so that they can be used for future research development.

Validity of the findings

(41) To end your abstract you can add a brief summary of the results obtained from this study. This provides a brief overview of the results of the study.
(969) You can explain the best results of GWO-ABE that are most accurate on which dataset, and the results that are less accurate than other methods also on which dataset. This is a reference in using different datasets in the future or improving GWO-ABE with the same dataset.

---

## Round 0.4 · accepted · Accept

Dear Authors,

Thank you for clearly addressing the reviewers' comments. Your paper seems sufficiently improved. I think your paper seems ready for publication.

Best wishes,

·

Basic reporting

no comment

Experimental design

no comment

Validity of the findings

no comment

Additional comments

The quality of the revised manuscript has been improved and complete. Then, the manuscript can be accepted for publication.